# Predictors of genomic differentiation within a hybrid taxon

Angélica Cuevas[1]*, Fabrice Eroukhmanoff[1], Mark Ravinet[1,2], Glenn-Peter Sætre[1], Anna Runemark[3]

1 Department of Biosciences, Centre for Ecological and Evolutionary Synthesis, University of Oslo, Oslo, Norway, 2 School of Life Sciences, University of Nottingham, Nottingham, United Kingdom, 3 Department of Biology, Lund University, Lund, Sweden

* a.m.c.pulido@ibv.uio.no, angelica.cuevas@evobio.eu

## Abstract

Hybridization is increasingly recognized as an important evolutionary force. Novel genetic methods now enable us to address how the genomes of parental species are combined in hybrid lineages. However, we still do not know the relative importance of admixed proportions, genome architecture and local selection in shaping hybrid genomes. Here, we take advantage of the genetically divergent island populations of Italian sparrow on Crete, Corsica and Sicily to investigate the predictors of genomic variation within a hybrid taxon. We test if differentiation is affected by recombination rate, selection, or variation in ancestry proportions. We find that the relationship between recombination rate and differentiation is less pronounced within hybrid lineages than between the parent species, as expected if purging of minor parent ancestry in low recombination regions reduces the variation available for differentiation. In addition, we find that differentiation between islands is correlated with differences in signatures of selection in two out of three comparisons. Signatures of selection within islands are correlated across all islands, suggesting that shared selection may mould genomic differentiation. The best predictor of strong differentiation within islands is the degree of differentiation from house sparrow, and hence loci with Spanish sparrow ancestry may vary more freely. Jointly, this suggests that constraints and selection interact in shaping the genomic landscape of differentiation in this hybrid species.

## Author summary

Genomes of hybrid lineages are mosaics of those of their parent species and harbour variation that has the potential to facilitate adaptation when hybrids encounter diverse environments. However, genetic incompatibilities between parental species can also act to limit possible combinations of parental alleles, constraining hybrid genome formation. What is the relative importance of selection and constraints in form of admixture proportions and genomic architecture in this process? We investigated this in the Italian sparrow, a hybrid species resulting from past hybridization between the house and Spanish sparrow. Using three independent hybrid lineages, we addressed how their genomes, harbouring different parental combinations, have evolved. We examined the roles of selection due

**Data Availability Statement:** Genomic data is deposited at the NCBI Sequence Read Archive under the BioProject ID PRJNA766344, BioSamples accessions SRR16763187-SRR16762873 and SRR16073354 -

SRR16073045. Other data including vcf files and intermediate analyses files are deposited in the Dryad Digital Repository at https://doi.org/10.5061/dryad.wpzgmsbns. And scripts developed in this study can be found in the Zenodo Repository at https://doi.org/10.5281/zenodo.5928415.

**Funding:** This research was funded by a Wenner-Gren fellowship to A.R., a Norwegian Research Council grant to G-P.S. and A.R., and a Marie-Curie Fellowship (2011-302504) and a University of Oslo, Faculty of Sciences grant to F.E. The funders had no role in study design, data collection and analysis, decision to publish, or preparation of the manuscript.

**Competing interests:** The authors declare that no competing interests exist.

to divergent local adaptation, recombination and purging of genetic incompatibilities in predicting differentiation. We found that selection against incompatibilities may constrain hybrid genome composition. In addition, signals of local selection as well as estimates of differentiation were correlated across populations, and outliers were shared among the hybrid lineages more often than expected by chance. Overall, our results suggest that in the Italian sparrow selection interacts with constraints linked to genetic incompatibilities affecting which sections of the genome can readily diverge among hybrid lineages.

## Introduction

Heritable variation is the substrate on which natural selection acts, and hybridization is increasingly recognized as an important process providing such variation in fish [1,2], insects [3], birds [4,5] and even humans [6]. Hybridization can enable lineages to combine parental genomes in adaptive ways, for instance contributing alleles linked to insecticide resistance in mosquitoes malaria vectors [7], adaptive fur colour in hares [8] and MHC immune defence diversity in modern humans [9]. Similarly, the variation created by hybridization has provided the raw materials for the extensive adaptive radiations of African lake cichlids [2,10–13]. Lineages resulting from hybridization may even outcompete the parental species in certain environments and colonize new niches as documented in *Helianthus* sun flowers where hybrid taxa colonize extreme environments [14,15]. One outcome of hybridization is hybrid speciation, resulting in the formation of a taxon that is reproductively isolated from its parent species [16]. Hybrid speciation can arise both through allopolyploidization and homoploid hybrid speciation, without an increase in chromosome number in the latter case [16–19]. Interestingly, the relative contributions of the parental species may vary within a hybrid, as illustrated by the variable genome composition in sword-tail guppies [20], in Lycaides butterflies [21], and, as is the focus of this study, in isolated island populations of Italian sparrows [22]. Here, we focus on the Italian sparrow, a well known example of a homoploid hybrid species, with reproductive barriers to the parent species consisting of a subset of those isolating the parent species [23,24]. Genetically divergent island populations of the Italian sparrow, potentially originating from independent hybridization events, differ in proportions of their genomes inherited from their parental species, house and Spanish sparrows (*P. domesticus* and *P. hispaniolensis*) [22]. The share of house sparrow ancestry, estimated as admixture proportion, ranges from 37% in the lineage on Sicily, to 62% in Corsica and 76% in the Cretan lineage [22].

Hybridization, was, at least in animals, historically viewed as an evolutionary mistake [25], partly because hybrids are likely to suffer from incompatible allelic combinations. While this view has changed over the last decades [17,26], hybrid lineages likely need to overcome a number of challenges to successfully establish. Incompatibilities might mean low fertility, sterility or even inviability in some crosses [27]. This is shown by Haldane's rule [28], when species have heterogametic sex chromosomes, the heterogametic sex is more likely to be sterile or inviable. In addition, evidence for a role of mito-nuclear interactions causing fitness reduction in hybrids is mounting [23,29,30]. For example, maladaptive metabolisms in hybrids [31] suggest that mito-nuclear interactions could pose strong selection pressures on the genomic composition in hybrid taxa. Mito-nuclear interactions may also play a role in determining the Italian sparrow genome composition [23]; hybrid Italian populations are largely fixed for the house sparrow mitochondrial genome, and there is evidence of an excess of house sparrow ancestry conserved in nuclear genome regions contributing to mitochondrion function [22]. Even in

species that have successfully formed hybrid daughter lineages, early generation hybrids may still be inviable or infertile [32]. These findings suggest that fitness losses due to incompatible parental combinations, i.e. Bateson-Dobzhansky-Muller incompatibilities (BDMI) [33–37], may be restored through fixation of compatible pairs of alleles from either of the parent species. Alternatively, if the portion of the genome that is free to vary (i.e. where constrains may be reduced) is reduced, it could potentially result in convergent allelic compositions at specific genomic regions among independent hybrid lineages.

Although hybrid lineages in principle have a vast number of potential combinations of parental alleles and increased nucleotide diversity available as a source for adaptation, little is known about genome stabilization in hybrid taxa [19,38]. After reproductive isolation from the parental species develops, stabilization of the hybrid genome will occur, removing ancestry blocks by purging of incompatibilities and fixing genomic combinations [39,40]. The speed of genomic stabilization varies between hybrid taxa and will also occur at different rates in different parts of the genome. It could take several hundred of generations for neutral loci [39] or occur very quickly in functionally important regions [40,41]. In addition to drift and selection, ancestry sorting during genome stabilization also has a determining effect on the composition of admixed genomes [42], which in turn affects patterns of genomic differentiation among hybrid populations. There could also potentially be constraints- here defined as effects of genomic architecture, including recombination rate, or incompatibilities due to ancestry admixture- on which genomic regions are free to vary and bias on the overall composition of hybrid genomes. For instance, introgression on the sex chromosomes is commonly reduced compared to genome-wide levels in species where one sex is heterogametic, consistent with selection against infertility [43–45]. Experimental assays in sunflowers and recent studies of *Lycaeides* butterflies have shown that the same genetic combinations found in natural hybrid lineages re-emerge in experimental hybrid populations [46] and younger lineages [47], possibly due to selection against alternative combinations and recombination effects. This raises the question of how easily hybrid lineages can achieve divergent genome compositions and phenotypes. Can different combinations of parental alleles easily be achieved due to selection for divergent local adaptation in homoploid hybrids? Or do patterns of differentiation at a local scale mirror those between strongly divergent populations, suggesting a role for constraints from recombination rate on which genome regions may differentiate? Exploring the patterns of population differentiation within hybrid species may reveal novel insights into the forces shaping hybrid genomes.

Interestingly, patterns of species differentiation are affected by the recombination rate landscape [48–50]. This can result in highly correlated patterns of divergence between closely related species pairs, such as that found in flycatchers [48]. Moreover, linked selection has a greater effect in regions of low recombination [48]. Selective sweeps in genomic regions of low recombination can give rise to a negative correlation between recombination rate and genomic differentiation [48,51]. Specific to hybrid taxa, evolutionary processes occurring during genome stabilization could have an impact on the distribution of genome diversity and later potential differentiation between independent hybrid populations [19,40]. For example, initial ancestry sorting could lead to differences in admixture proportions. This could give rise to subsequent lineage specific evolution within ancestry types. Purging of minor parent ancestry in low recombination regions to reduce genomic incompatibilities could reduce the variation available for subsequent differentiation. Recent studies have indeed found reduced introgression in low recombination regions in hybrid swordtail fish, sticklebacks, *Heliconius* butterflies and humans, suggesting that the recombination landscape may indeed affect which regions are permeable to introgression [20,49,52–54]. Genomic blocks with ancestry from the minor-parent may be retained in regions of high recombination rate, due to their increased likelihood

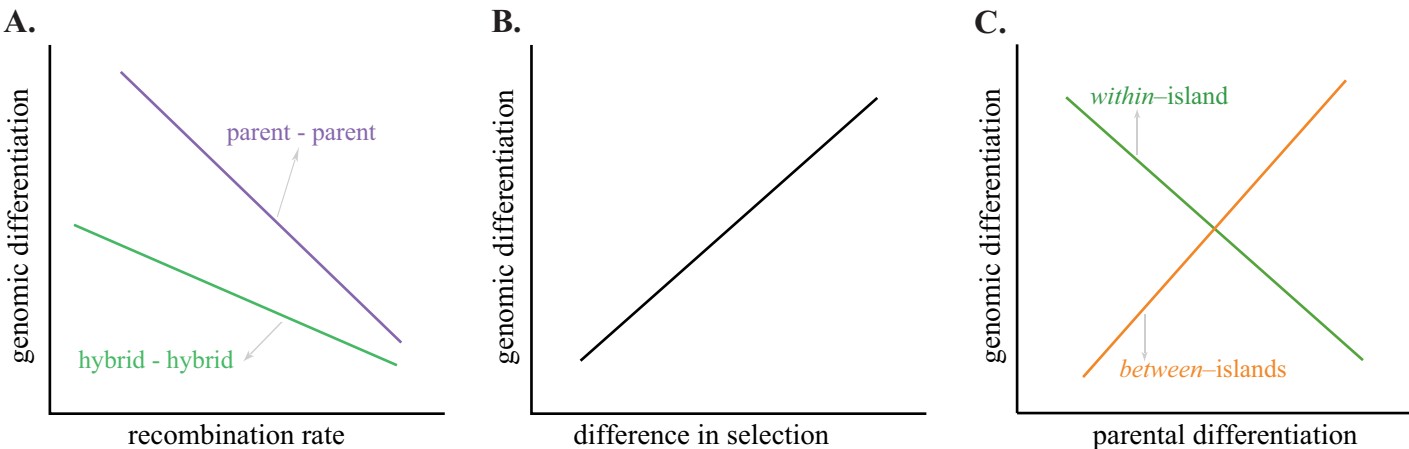

**Fig 1. Expected patterns of genomic differentiation for different levels of recombination rates, divergent selection and parental differentiation. A. Recombination rate.** *Parent-parent*: Since selective sweeps have a greater effect in low recombination regions we expect a negative relationship between recombination rate and differentiation between the non-hybrid parental lineages. *Hybrid—hybrid*: Specific to hybrid taxa, additional selection against minor parent ancestry in low recombination regions could reduce the variation available for differentiation in these regions. As this process does not affect parent taxa, we would expect a flatter relationship between recombination rate and differentiation in hybrids, compared to their parent taxa, if purging of minor parent ancestry is pervasive. In addition, in higher recombination regions there is a potential for alternative blocks of ancestry to be fixed, which will inflate the differentiation among hybrids relative to the parent-parent comparison, contributing to the flatter relationship in hybrids **B. Selection.** If divergent selection is an important predictor of differentiation, we expect differentiation between two islands to be positively correlated to the strength of divergent selection between them. **C. Parental differentiation.** If differentiation is limited by potential incompatibilities between divergent parental loci, we would expect the potential for adaptive differentiation in hybrid populations of similar origin to be highest in genomic regions of low parental divergence. Regions that are strongly differentiated between parents, more likely to be incompatible in combination with other parental alleles, would have a greater likelihood of being fixed for the ancestry of one parent only during genome stabilization. This process is expected to act *within islands*, where the overall genomic composition and the resolution of incompatibilities through fixation of alleles from one parent species, are most likely similar. In contrast, in independent hybrid populations with different proportions of ancestry (like the ones found in different islands), fixation of alternative parental alleles could increase the potential for differentiation and local adaptation *between islands*. These processes, with influence from processes not addressed in this study, affect the genomic composition in hybrid lineages. However, their relative importance may be inferred from the signatures of differentiation in relation to the predictors.

of breaking away from potential incompatibilities in these regions [20]. If, during genome stabilization, purging of incompatibilities in low recombination regions is pervasive, the resulting reduced diversity in these regions could also act as a constraint on genomic differentiation in the hybrid, decreasing differentiation at a greater rate in low- than high recombination regions. Thus, the relationship between recombination rate and differentiation is expected to be less negative in hybrid lineages compared to the differentiation between the non-hybrid parental taxa should such purging be important (Fig 1A). In contrast, local selective sweeps, having a greater impact on regions of low recombination due to linked selection, could lead to higher differentiation in low recombination regions in comparison to regions of high recombination. However, the relative importance of these two processes in shaping hybrid genomes remains unclear.

There is evidence that hybrid taxa can use the variation that originated through hybridization for local adaptation. For instance, beak shape in Italian sparrows is explained by local precipitation regime [55]. Beak size differences among island populations are best explained by temperature seasonality [56], and some island populations are strongly differentiated for a gene known to affect beak morphology in Darwin's Finches, FGF10 [22,57]. In addition, the gene GDF5, part of the BMP gene family that has a fundamental role in beak shape and size variation in Darwin's finches [58,59], is a candidate gene putatively under selection in the Italian sparrow populations from mainland Italy [38]. However, beak shape is also affected by the proportion of the genome inherited from each parent species suggesting a small albeit significant role for admixture proportions in morphology [55,56]. Moreover, in a recent study

investigating genetic differentiation among Italian sparrow populations on mainland Italy, population differentiation was associated with temperature and precipitation [38]. Interestingly, alleles segregating in both parental species showed strong allele frequency differences within the Italian sparrow, suggesting that adaptation is not completely dependent on the combination of alleles from different parent species [38]. If selection has a significant role in genomic differentiation of the Italian sparrow and selection pressures diverge among islands, favouring the fixation of alternate alleles, we would expect stronger differentiation in regions under divergent selection among islands (Fig 1B). Furthermore, depending on the variation of parental ancestry proportions in hybrid populations, parental divergence may affect genomic differentiation in the hybrid in different ways. In populations with similar parental ancestry, like those within islands, the resolution of incompatibilities during genome stabilization is likely to be similar. For these, genomic differentiation may be most easily achieved from standing genetic variation inherited from the parents, thus from variants that are segregating in the parent species (Fig 1C). Alternatively, in independent hybrid populations with divergent proportions of parental ancestry, differentiation is more likely to be found in regions of strong parental differentiation (Fig 1C).

One way to further our understanding of the evolutionary forces acting on hybrid genomes is to investigate patterns of differentiation within hybrid lineages and the factors that best explain them. The Italian sparrow is a uniquely suited study system, as it provides independent populations with varying ancestry proportions (Fig 2). These island lineages are likely to have originated as a result of different hybridizations events [22]. This is supported by the low pairwise correlations of ancestry tracts among islands and significant albeit small differences in ancestry tract sizes [22], suggesting at least long periods of independent evolution. This unique system enables comparison of hybrid lineages with divergent ancestry proportions, as well as comparison of populations with potentially similar parental contributions i.e., populations within islands with similar evolutionary history. Here, we use the island Italian sparrows to investigate how differentiation within island compares to that among islands. Our overachieving aim is to address which factors best predict differentiation within and among islands to disentangle the evolutionary forces shaping hybrid genomes.

We test the hypotheses that I) long periods of independent evolution have resulted in significantly higher divergence *among*-islands than *within*-islands; II) selective sweeps and purging of minor parent ancestry in low recombination regions in the hybrid has led to a less steep relationship between recombination rate and differentiation than that between the parent species (Fig 1A) III) that genomic regions experiencing stronger divergent selection *among* islands also show elevated differentiation (Fig 1B) IV) that constraints on how freely genomic regions are able to diverge have led to correlated landscapes of differentiation *within*- and *among*-islands, and V) that differences in minor–major parental ancestry in the hybrid have a direct effect on the genomic differentiation between populations.

## Results

### I) Genomic differentiation within- and between islands

Consistent with [22] we find strong differentiation between the focal island populations based on RAD data. From our principal component analysis, based on 2224 SNPs (S2 Table), the first main axis of differentiation largely reflects the proportion of the genome inherited from each parent species, and Crete diverges along the second axis (Fig 2C). Interestingly, the ADMIXTURE analysis supports the presence of three clusters rather than two, with Crete forming a separate cluster (Fig 2B). Average of windows-based $F_{ST}$ estimates, based on 2856 SNPs (S2 Table), are consistent with this, Crete is more strongly differentiated from Sicily

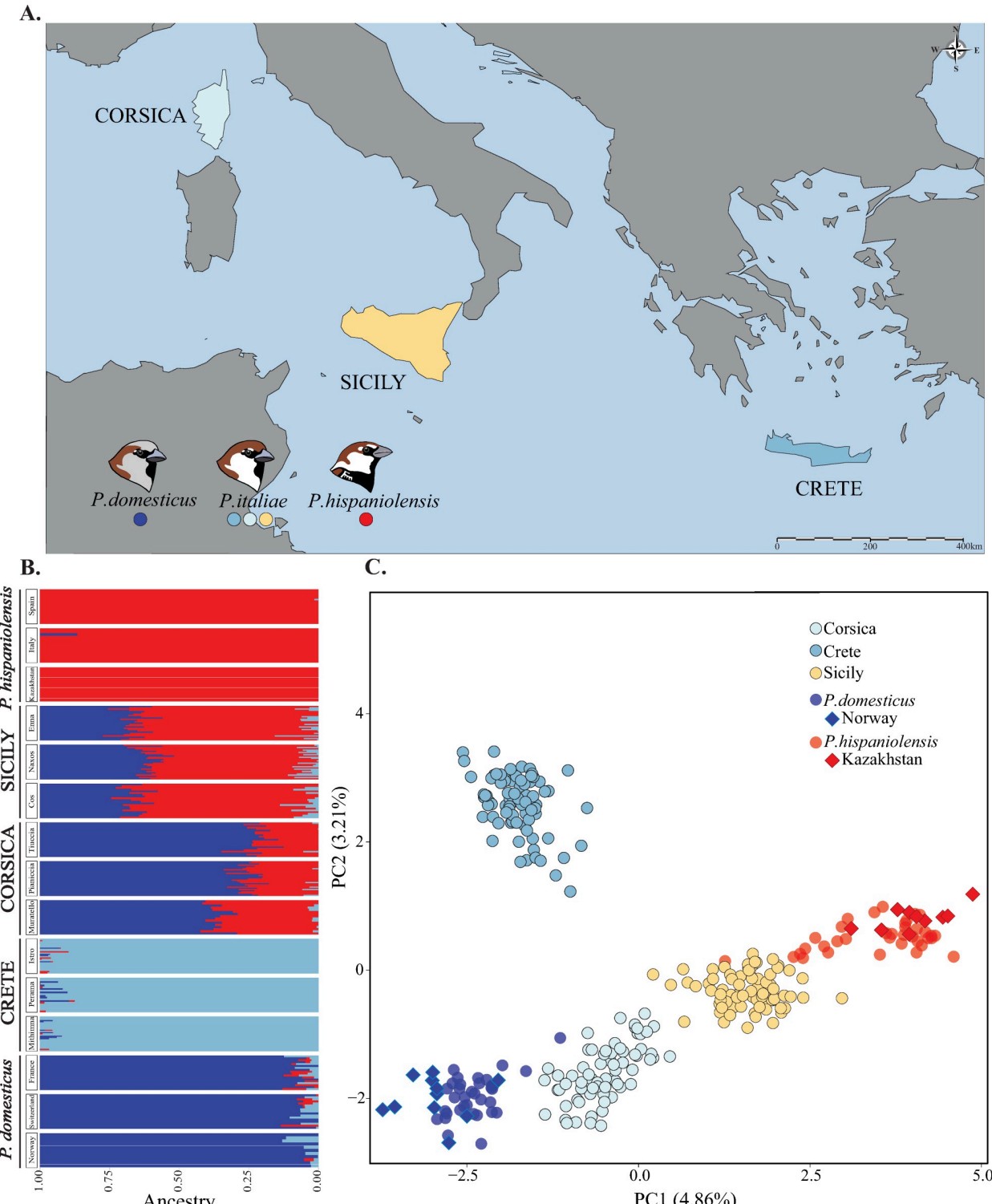

**Fig 2. Sampling design and population structure. A.** Map of sampling locations. Sparrows were sampled from three populations on each of the islands of Corsica (N = 70), Crete (N = 77) and Sicily (N = 76) **B.** Admixture analysis illustrating the clustering of the island populations of the Italian sparrow and their parental species for three clusters (K = 3), the value that received the highest support. Three geographically separated populations of each of the parental species, the house and Spanish sparrow, were included. **C.** Principal component analysis (PCA) illustrating the relationship between the Italian sparrow populations and the parental species. ◆ indicate the reference parental populations with lower levels of introgression. Analyses were based on a VCF containing 2224 SNPs. Map base layer was retrieved using the R-package "rworldmap" and the function getMap() - https://cran.r-project.org/web/packages/rworldmap/.

(mean $F_{ST}$ = 0.043) and Corsica (0.042), whereas mean $F_{ST}$ between Corsica and Sicily is comparatively lower (0.025; Fig 3A).

Differentiation is more pronounced between islands than within islands (Monte-Carlo permutation paired t-test: t = 33.21, df = 7927, $P$ = 1.98e-15; Table A in S1 Text; Fig 3A), with a medium effect size estimate (Cohen's d = 0.523; 95% CI 0.49–0.56). A discriminant function analysis, using the dataset of 2856 SNPs (S2 Table), recovers some differentiation among local populations in each island (Fig 3B and 3C), and correctly assigns 95.3% of Corsican individuals, 78.4% of the individuals from Crete and 75.5% of the Sicilian individuals to their populations of origin within each island. *Within*-island $F_{ST}$ differs significantly among islands (All $P$s < 0.5e-3; Table A in S1 Text), with Corsican populations exhibiting the highest mean $F_{ST}$ of 0.018, as well as the highest nucleotide diversity (π: 3.021e-06; Table A in S1 Text). Differentiation within Sicily is intermediate at 0.013, while $F_{ST}$ among the Cretan populations is the lowest at 0.011 (Fig 3A and Table A in S1 Text). While most variation segregates within individuals and populations, an AMOVA reveals that 4.84% of the variation is found among islands whereas 0.91% of the variation is found among populations within islands (both fractions are statistically significant $P$: 0.001, as estimated from a randomization Monte Carlo test with 1000 permutations; Table 1 and Fig A in S1 Text).

## II) The relationship between genomic differentiation and recombination rate

To evaluate the hypothesis that hybrid-specific purging of minor ancestry blocks in low recombination regions can reduce genomic diversity (and in turn genomic differentiation) in these regions in contrast to high recombination regions, where the effect of purging is expected to be weaker; we tested if genomic differentiation decreases less rapidly with recombination rate between populations within islands than between the parent species (Fig 1A). We evaluated this by comparing the slopes of the relationship between recombination rate and differentiation and through testing for a significant interaction effect between the type of comparison (parent-parent vs. *within*–island) and recombination rate on differentiation. Using differentiation among populations within islands implies that relatively similar resolution of incompatibilities across populations can be assumed. We predicted that the slopes would be less steep among populations from the same island (*within*-island $F_{ST}$) than between the parent species, if selection against minor parent ancestry is an important selection pressure in the hybrid lineages, reducing variation in low recombination regions (Fig 1A). Indeed, we find that this is the case (Tables B and C in S1 Text; Fig 4A). The slope generated by the relationship between the parental differentiation and recombination rate differs from those found in each *within*-island comparison (Table B in S1 Text; Fig 4A). We find a significant interaction of recombination rate and comparison (parent-parent vs. within-island) in all independent linear models per island (Table C in S1 Text). We find no significant correlation between recombination rate and *within*-island genomic differentiation for Corsica (correlation = -0.012, $R^2$ = 1.4e-4, $P$ = 1; Fig 4B) or Crete (correlation = -0.003, $R^2$ = 0.9e-5, $P$ = 1; Fig 4C). However, differentiation within Sicily is weakly but significantly negatively correlated with recombination rate, albeit the effect size is very small (correlation = -0.048, $R^2$ = 0.002, $P$ = 0.037; Fig 4D).

When evaluating the influence of recombination rate, parental differentiation and differentiation to the two parent species, a GLM did not reveal any significant relationship between recombination rate and differentiation within islands (GLM, Estimate = -9.66e-04, Std. Error = 7.84e-04, $P$ = 0.22; Table D in S1 Text). Corresponding binomial models revealed that recombination rate did not affect the probability of loci being $F_{ST}$ outliers (1% outliers of the $F_{ST}$ distribution) within islands either (Table 2, Table E in S1 Text). However, evaluating

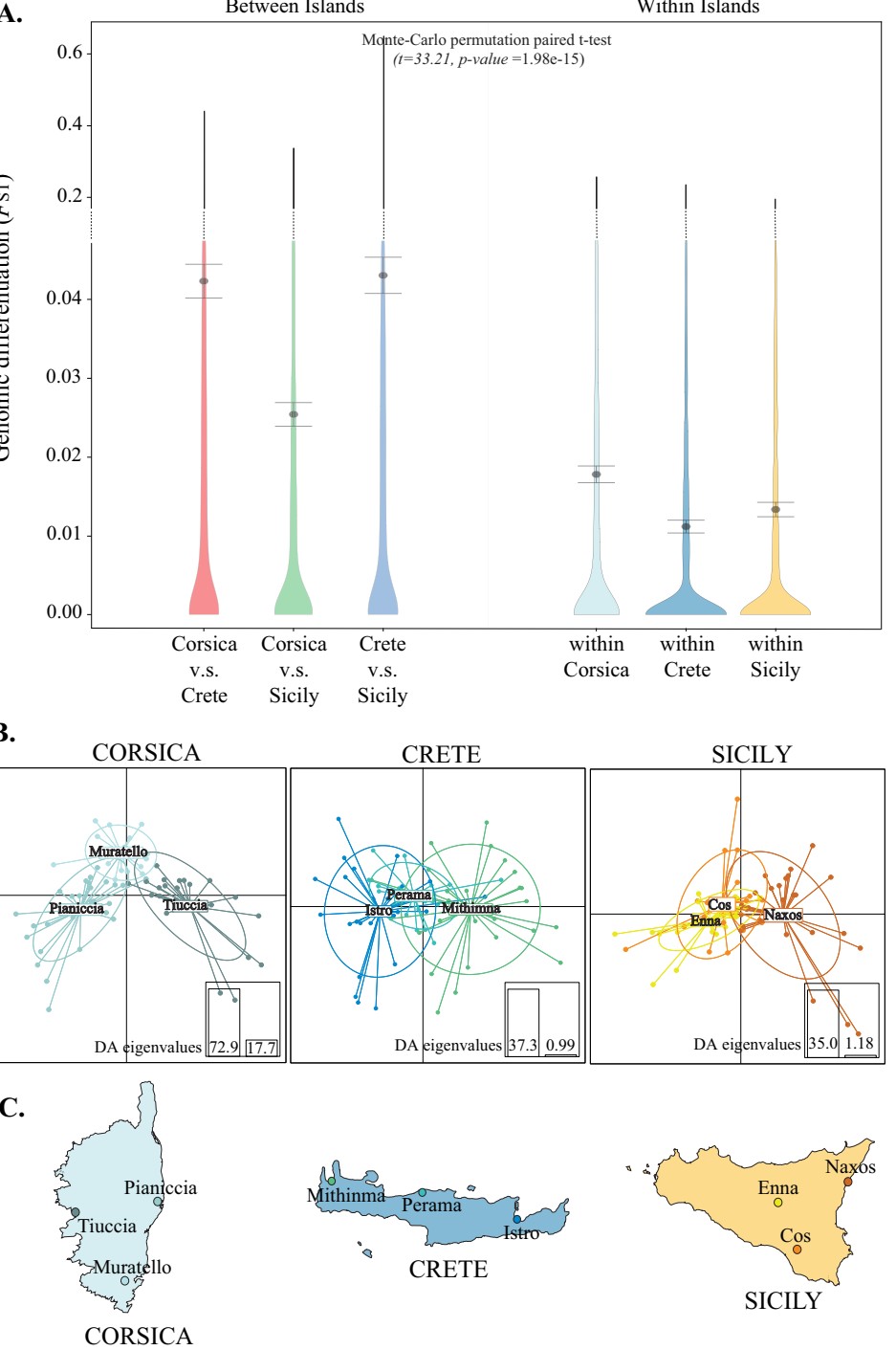

**Fig 3. Genomic differentiation *within*- and *among*-islands. A.** Levels of genomic differentiation among the three islands (window based $F_{ST}$ between islands pairs) compared to levels of differentiation within islands (window based $F_{ST}$). Points and bars depict the means and 95% CI, respectively. Continuity of the y-axis is broken (dashed line) to minimize the size of the figure in order to include the extreme values of the $F_{ST}$ distribution. **B.** Discriminant analysis of principal components (DAPC) for populations within Corsica, Crete and Sicily, respectively. **C.** Map of the focal islands with the three populations sampled within each island denoted by filled circles. All analyses were based on a VCF containing 2856 SNPs. Maps base layer was retrieved using the R-package "rworldmap" and the function getMap (). - https://cran.r-project.org/web/packages/rworldmap/.

**Table 1. Analysis of Molecular Variance (AMOVA) across islands and populations within islands.** Several cut-offs for missing-ness per loci were used (5%, 10%, 20%), but the results from the AMOVA did not change substantially. Here we present the results from a cut-off of 5% (Table L in S1 Text). The significance of variation partitioning in each element was maintained. Analyses are based on a VCF containing 2856 SNPs.

| Analysis of Molecular Variance | | | | | Randomization by Permutation | |
|---|---|---|---|---|---|---|
| AMOVA | | | | | Monte Carlo test | |
| Variance partitioning | Df | Sum Sq | Sigma | % of covariance | Std. Observed | *P-value* |
| Among islands | 2 | 778.79 | 2.22 | 4.84 | 4.33 | 0.001** |
| Between populations within island | 6 | 365.37 | 0.42 | 0.91 | 9.42 | 0.001** |
| Between individuals within populations | 213 | 8654.88 | -2.54 | -5.54 | -4.17 | 1.000 |
| Within individuals | 222 | 10148.04 | 45.71 | 99.80 | 1.26 | 0.897 |

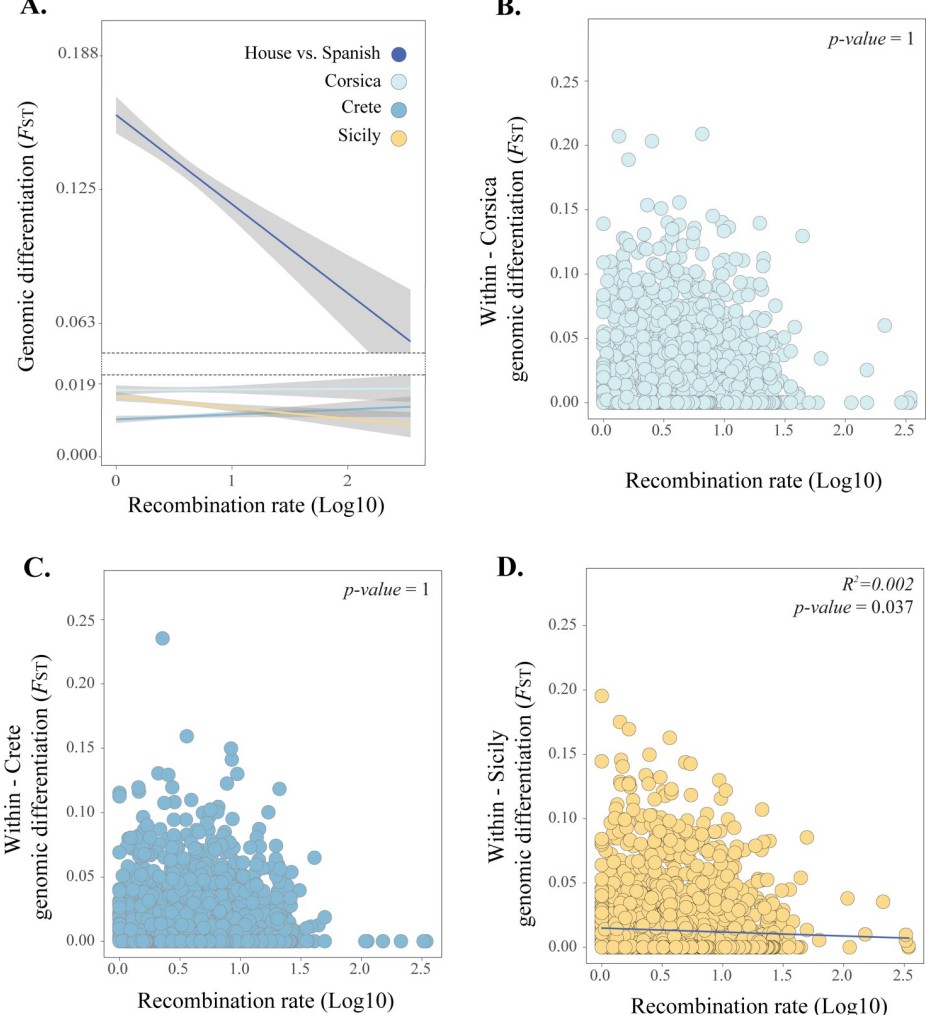

**Fig 4. The influence of recombination rate on genomic differentiation. A.** Comparison of the relationship between recombination rate and genomic differentiation between the parental species (dark blue) and between Italian sparrow populations within each island. Correlation between recombination rate and genomic differentiation within island for **B**. Corsica, **C**. Crete and **D**. Sicily. Analyses based on a VCF containing 2804 SNPs for **A** and one containing 2856 SNPs for **B, C** and **D**.

**Table 2. Logistic regression assessing the predictors on the probability of being a *within*–island $F_{ST}$ outlier and post-hoc estimated marginal (least-square) means.**

| Model: | | | | |
|---|---|---|---|---|
| Pr(*within*-island $F_{ST}$ outlier) = per-loci local ancestry proportion (LLAP) + Recombination Rate + island + island.house $F_{ST}$ + island.Spanish $F_{ST}$ + house.Spanish $F_{ST}$ | | | | |
| **Response variable** | **Predictor** | **Estimate** | **Std. Error** | ***P-value*** |
| Pr (*within*-island $F_{ST}$ outlier) | Recombination Rate | -1.402e-01 | 1.431e-01 | 0.33 |
| | LLAP | 1.411e-02 | 6.336e-02 | 0.82 |
| | Island v.s House $F_{ST}$ | -1.401e+00 | 6.674e-01 | 0.04* |
| | Island v.s Spanish $F_{ST}$ | 5.728e-01 | 4.698e-01 | 0.22 |
| | House v.s Spanish $F_{ST}$ | -6.752e-06 | 8.770e-06 | 0.44 |
| **Post-hoc Estimated marginal (Least-squares) means** for the predictor variable "island" | | | | |
| **Comparison** | **Estimate** | **Std. Error** | **Z ratio** | ***P-value*** |
| Corsica—Crete | 0.021 | 0.133 | 0.16 | 0.99 |
| Corsica—Sicily | -0.11 | 0.133 | -0.88 | 0.67 |
| Crete—Sicily | -0.13 | 0.134 | -1.03 | 0.58 |
| *P-value* adjustment: Tukey's HSD | | | | |

individual islands separately shows that recombination rate significantly explains differentiation between Sicilian populations, with higher divergence in low recombination regions, as revealed by a GLM (Parameter estimate = -3.48e-03, Std. Error = 1.34e-03, *P* = 9.3e-03; Fig 4D; Table F in S1 Text).

## III) The concordance of patterns of selection and genomic differentiation

To assess the role of selection in shaping genomic differentiation in the Italian sparrow, we tested if differences in selection were correlated to genetic differentiation. If divergent selective pressures among the islands have a large influence on the formation of their hybrid genomes, we expect a positive correlation between measures of divergent selection and genomic differentiation between islands (Fig 1B). Here, we tested this prediction, using cross population haplotype homozygosity (xp-EHH) a statistic that measures putative patterns of divergent selection by comparing haplotype lengths between populations to detect potential selective sweeps [60]. Genomic differentiation between two island pairs was significantly correlated to xp-EHH. This measure was negatively correlated with $F_{ST}$ for the Corsica–Sicily (correlation = -0.061, $R^2$ = 0.0037, *P* = 0.014), and Crete–Sicily (correlation = -0.059, $R^2$ = 0.004, *P* = 0.019; Fig 5A; Table G1 in S1 Text) comparisons. However, the effect of xp-EHH on differentiation between these islands was very small with an almost marginal effect size, and there was no relationship between xp-EHH estimates and differentiation in the Corsica–Crete comparison (correlation = -0.042, $R^2$ = 0.002, *P* = 0.148; Fig 5A).

We also addressed how consistent selection is within- and across islands. Patterns of selection within islands (estimated by the integrated haplotype homozygosity score, iHS) were positively correlated in all pairwise comparisons between islands, with $R^2$ ranging from 0.095 to 0.174 (Fig 5B; Table G3 in S1 Text). This suggests shared patterns of selection across the genomes, potentially driven by similar selection pressures or genomic constraints arising from the distribution of variation and incompatibilities in the parent species, reducing the availability of genomic variation. However, differentiation within island populations was not significantly correlated to iHS for neither Crete nor Corsica (Fig B and Table G2 in S1 Text), although there was a weak correlation with a small effect size for Sicily ($R^2$ = 0.005, *P* = 0.01; Fig B and Table G2 in S1 Text). Interestingly, mean Tajima's D differed considerably among islands, with Sicilian and Corsican populations exhibiting negative estimates (range from -0.25

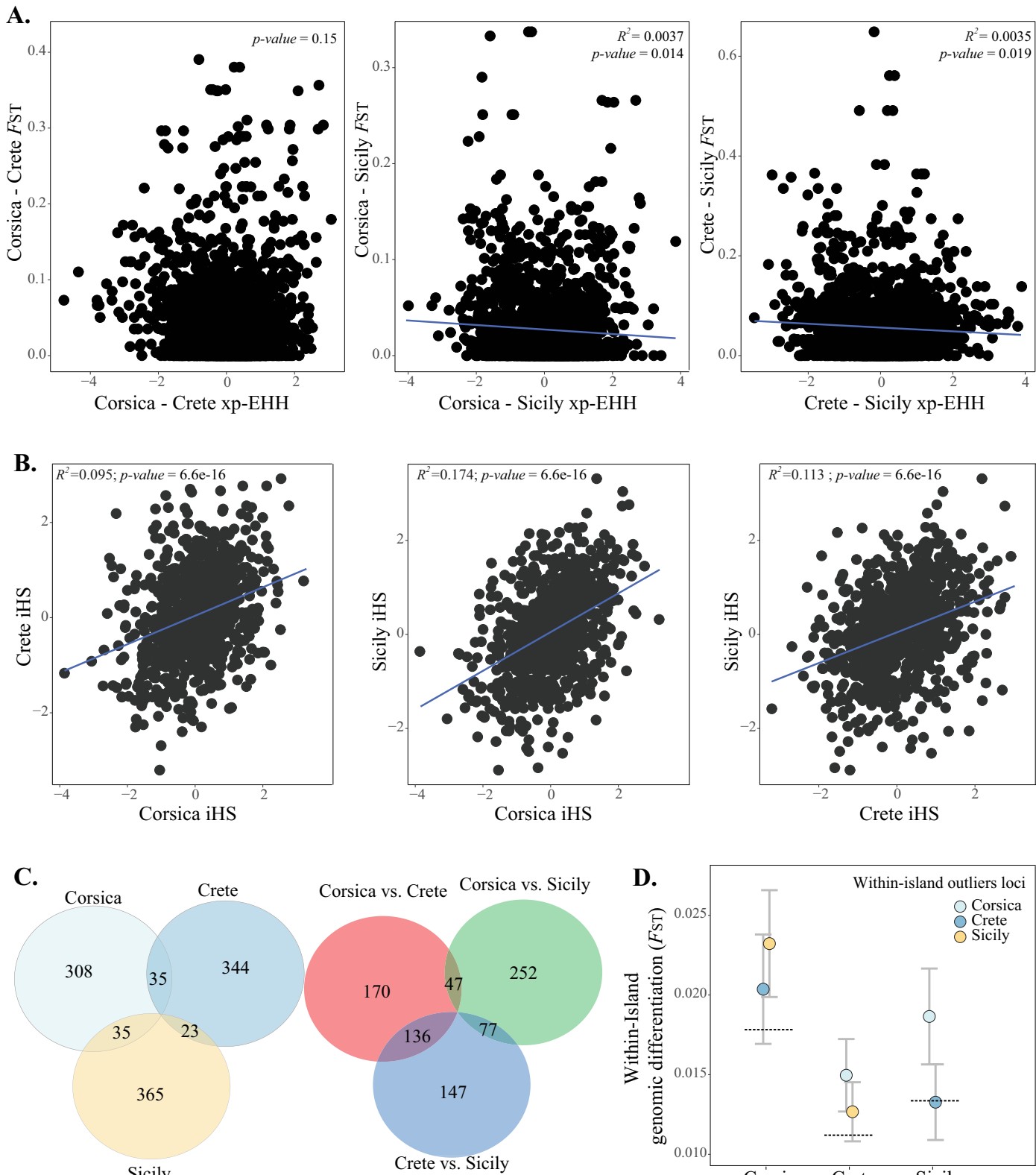

**Fig 5. Effects of selection on genomic differentiation, similarity in selection pressures across islands and shared patterns of genomic differentiation. A.** Relationship between divergent selection (estimated as Extended Haplotype Homozygosity statistic—XP-EHH) and genomic differentiation between islands. **B. Similarity in patterns of selection.** Between islands correlations of the *within*-island selection measure, estimated as integrated haplotype homozygosity score

(iHS). **C.** Shared outlier loci among populations within islands (left) and among islands (right). **D.** $F_{ST}$ for outlier loci from each of the other islands within each island. Dashed lines represent the global within-island $F_{ST}$ mean. Error bars denote 95% CI.

to -0.048; Fig C and Table A in S1 Text). In contrast, populations on Crete exhibited higher values of Tajima's D (range from -0.11 to 0.015; Fig C and Table A in S1 Text).

## IV) Distribution and repeatability of differentiation across the genome

To evaluate concordance in the differentiation landscape, as expected if genomic differentiation is affected by recombination rate, constraints or similar selective pressures, background or parallel selection, we ran correlations of *within*-islands and *among*-islands $F_{ST}$. We found that patterns of *within*-island differentiation are significantly correlated between Sicily and Corsica (correlation = 0.081, $R^2$: 0.0066, *P* = 1.16e-10), but not between Corsica and Crete (correlation = 0.02, $R^2$: 5.8e-4, *P* = 0.16), or Sicily and Crete (correlation = 0.014, $R^2$: 1.96e-4, *P* = 0.81; Fig D in S1 Text). Hence, levels of differentiation are not correlated between all islands. Interestingly, we found that the outlier loci within one island were more frequently outliers within other islands than expected by chance in two out of three comparisons (Fig 5C; Table H in S1 Text). A total of 9.3% of the outliers within Corsica overlap with those from Crete (Chi-squared: 7.18, *P*: 0.007) and a similar percentage (9.3%) in within–Sicily $F_{ST}$ outliers (Chi-squared: 6.80, *P*: 0.009; Table H in S1 Text). However, outliers from Crete and Sicily are not shared to a higher extent than expected by chance (Chi-squared: 0.09, *P*: 0.767). We also tested whether individual island outliers have a higher mean $F_{ST}$ within other islands. We found outliers from Sicily to have a higher $F_{ST}$ values within Crete and within Corsica than expected by chance (Crete: t = -1.997, df = 457.9, *P* = 0.046; Corsica: t = -3.082, df = 444.7, *P* = 0.002; Fig 5D). Similarly, Corsica $F_{ST}$ outlier loci have elevated $F_{ST}$ within Sicily (t = -3.393, df = 392.2, *P* = 7.6e-4) and within Crete (t = -3.586, df = 385.1, *P* = 3.8e-4). However, outliers from Crete do not have higher $F_{ST}$ values than expected by chance in any of the other island populations (Fig 5D).

We further tested whether loci differentiated within islands also are more differentiated among islands. Pair-wise correlations between *within*– and *among*–islands $F_{ST}$ suggest the same regions are differentiated, but the effect is weak and varied. While differentiation within Corsica is correlated to Corsica–Sicily $F_{ST}$ (correlation = 0.05, $R^2$: 0.0025, *P*: 0.013) and Crete–Sicily $F_{ST}$ (correlation = 0.05, $R^2$: 0.0025, *P*: 0.026; Fig E in S1 Text), none of the other seven comparisons are significant. Consistent with this pattern, the proportion of *within*–Corsica outlier loci that overlap with the most differentiated loci in Corsica–Sicily $F_{ST}$ (9.8%) and in Crete–Sicily $F_{ST}$ (7.4%) are also higher than expected by chance (Chi-square tests: $X^2$: 15.53, *P*: 8.1e-05 and $X^2$: 4.09, *P*: 0.04, respectively; Table I in S1 Text). Moreover, we found a higher proportion (10.2%) of Crete's outlier loci than expected among the Crete–Sicily $F_{ST}$ outlier loci ($X^2$: 21.13, *P*: 4.3e-06, Table I in S1 Text). Among the 56 putative genes located in the shared regions of differentiation there is one presenting mitochondrial functions (S3 Table).

To evaluate whether background selection or adaptive parallel selection shape the patterns of shared differentiation, we tested correlations across all possible pair-wise comparisons of subpopulations within each island and compared these to all pair-wise correlations between populations on different islands. The rationale for this is that background selection should result in significant correlations in all analyses, as the correlations would reflect a conserved recombination rate landscape resulting in elevated differentiation in low recombination regions. We find variation in the strength of the relationships depending on comparison, with stronger relationships between differentiation in some comparisons, including some border-line significant ones (Figs F and G and Table J in S1 Text).

## V) Patterns of local genomic differentiation in relation to parental contributions to the genome

Multiple factors may affect which loci are free to vary within the Italian sparrow. For example, variation in parental contributions to the genomes of the different island populations, the level of differentiation between the parent species across the genome, and the recombination rate. We tested to what extent these factors explain the patterns of *within*-island differentiation by performing a generalized linear model (GLM) and a logistic model using *within*-island $F_{ST}$ as the response variable. The factor that best predicts the probability of a SNP to be an $F_{ST}$ outlier within islands is the extent of differentiation to the house sparrow (Logistic regression estimate: -1.401e+00, *P*: 0.036; Table 2). However, when evaluating factors that may affect the *within*-island differentiation ($F_{ST}$), using a GLM, differentiation to the house sparrow was found to be non-significant (Table D in S1 Text). Neither the extent of differentiation from the Spanish sparrow, parental differentiation, recombination rate, nor the per-locus local ancestry proportion (LLAP) contributed significantly to differentiation in either, the logistic regression or the GLM (Table 2; Table D in S1 Text). Furthermore, in separate logistic regressions for each island, none of the studied factors significantly affected the probability of being an outlier (Table E in S1 Text), potentially because of reduced statistical power. However, in separate GLMs run for each island (Table F in S1 Text) including all the predictors mentioned above, the general distribution of differentiation ($F_{ST}$) within Corsica is explained by the differentiation to the Spanish sparrow (Estimate for Spanish $F_{ST}$ = 1.166e-02, *P* = 0.036, Multiple $R^2$ of the model = 0.003) and differentiation within Sicily is weakly but significantly negative correlated to recombination rate (Estimate for recombination rate = -3.475e-03, *P* = 9.3e-3, Multiple $R^2$ of the model = 0.004; Table F in S1 Text).

Parental contributions to ancestry differ among islands. For example, the Spanish sparrow is the minor-ancestry parent to Corsican and Cretan populations, while the house sparrow is the minor-ancestry parent for the Sicilian populations. Taking advantage of this variation, we addressed whether *within*-islands differentiation was correlated to the differentiation between the focal island and their minor-ancestry parent species. For Sicily and Corsica, there is a significant correlation between *within*-island differentiation and differentiation to their minor-ancestry parent; the house and Spanish sparrow, respectively (Corsica: $R^2$ = 0.002, *P*: 3.58e-4; Sicily $R^2$ = 0.001, *P*: 0.022; Fig 6A) but not to the alternative parent species in either case (Fig H1 in S1 Text). Differentiation within Crete was not correlated to differentiation to any of the parental species (Fig 6A; Fig H1 in S1 Text). Similarly, there is a pattern where *within*-island outliers are more differentiated to the minor-parent than the genome-wide neutral expectation in two out of three islands (Fig 6B) as outliers within Crete and within Corsica are more differentiated from the Spanish sparrow than expected based on the overall genome-wide average. Conversely, in Sicily *within*-island outlier loci are not more differentiated to the minor-parent, the house sparrow (Fig 6B). We also find higher divergence of *within*-islands outlier loci (1% $F_{ST}$ outliers) to the minor- than to the major-ancestry parent. Outlier loci within Corsica were significantly more differentiated from the Spanish sparrow (the minor-parent) than from the house sparrow (t = -6.22, df = 519.4, *P* = 1.01e-09), as were outlier loci within Crete (t = -2.96, df = 668.2, *P* = 3.17e-3). Outliers within Sicily, where the house sparrow is the minor-ancestry parent, are significantly more differentiated from house sparrow than from Spanish sparrow than expected (t = 3.76, df = 679.8, *P* = 1.81e-4; Fig 6B).

The degree of genomic stabilization can affect the potential for genomic differentiation in hybrid lineages, as both parental alleles are expected to segregate in populations with genomes that are not stabilized. To evaluate whether island populations differ in the degree of genomic stabilization we estimated the rate of fixation of differentially fixed parental alleles. Crete

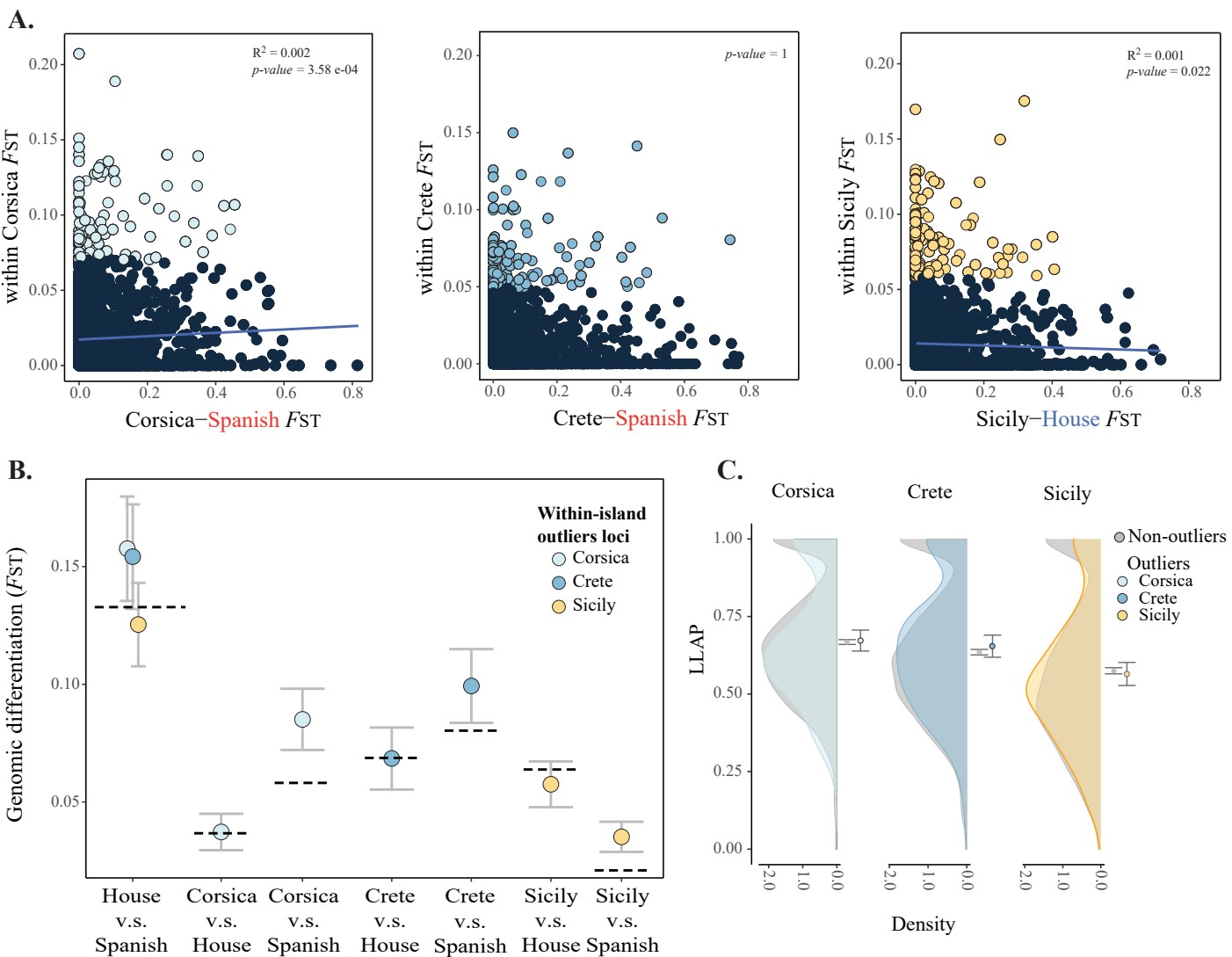

**Fig 6. Effects of divergence from parental species and minor−, major parent ancestry on differentiation. A.** Correlations of *within*-island $F_{ST}$ and genomic differentiation from the minor-ancestry parent (island vs. minor-parent $F_{ST}$). 1% $F_{ST}$ outliers are indicated in coloured dots in contrast to the non-outlier loci, in black. **B.** Parental differentiation (house-Spanish $F_{ST}$) and island-parent $F_{ST}$ for *within*-island outlier loci compared to the genome wide average (dashed line). **C.** Density of per locus local ancestry proportion (LLAP) for within-island $F_{ST}$ outliers, compared to the genome wide distribution. Means and 95% confident intervals of outlier and non-outlier loci are depicted.

shows the highest degree of fixation with a rate of 0.17, while Sicily has a fixation rate of 0.003 and Corsica with a negligible number of fixed loci parentally diverging. We further evaluated variation on the fixation rate of major and minor parental alleles independently, to address if there is evidence of preferential fixation of alleles from one of the parent species to reduce incompatible interactions. As expected, the ratio of fixed loci in the islands is higher for the ancestry from the major parent (Fig I in S1 Text). This pattern is found for two out of the three islands. While Crete has inherited more fixed loci from the house sparrow, Sicily present a higher ratio of fixed loci inherited from the Spanish sparrow. Hence loci from the major parent are more frequently fixed in these populations. Among the islands, Crete shows the highest rate of fixation of the major-parent ancestry (ratio = 0.168 for house sparrow ancestry), follow

by Sicily (ratio = 0.0014 for Spanish ancestry), while Corsica has an approximately equal number of fixed sites inherited from the two parent species (Fig I in S1 Text).

We also evaluated if divergence between the parental species (house-Spanish $F_{ST}$) affected *within*-island differentiation. Parental differentiation was weakly correlated to differentiation within Corsica (correlation = 0.058, $R^2$ = 3.4e-3, $P$ = 5.34e-5) but not to any other *within*-island differentiation (Fig H2 in S1 Text). Consistent with this, Corsica outlier loci also had higher parental differentiation than expected by chance (t = 2.15, $P$ = 0.033; Fig 6B), but this does not hold true for Crete (t = 1.852, $P$ = 0.065) or for Sicily (t = -0.811, $P$ = 0.42). Finally, differentiation *within*-Corsica and *within*-Sicily was not correlated to differentiation among populations within either of the parental species (Fig J in S1 Text) and differentiation within the parental species was not higher than expected by chance for outlier loci from these islands (Fig K in S1 Text). Interestingly, genome-wide differentiation within Crete was negatively correlated to differentiation within the house sparrow, and differentiation to the house sparrow was lower than expected by chance for outliers within Crete (Figs J and K in S1 Text). Jointly, these results suggest that outlier loci among populations within islands are not dependent on differentiation among populations within each of the parent species.

We assessed the effect of ancestry divergence on genomic differentiation within islands. We use a per locus local ancestry proportion (LLAP, estimated using whole genome sequencing data from [5], [22] and [61]). Whereas a LLAP of 0 corresponds to only Spanish ancestry, 1 corresponds to pure house sparrow ancestry. The distribution of the LLAP does not differ between the *within*-island $F_{ST}$ outlier and the non-outlier loci for any of the islands (t-tests with t = 0.27, 1.04 and -0.53, for Corsica, Crete and Sicily, respectively; P>0.05 for all islands; Fig 6C). A post hoc correlation analysis shows that *within*-island differentiation is not affected by ancestry ($R^2$ ranging from 5.1e-4 to 1.8e-5, with P>0.05 for all islands; Fig L1 in S1 Text). We further tested if the local ancestry (LLAP), estimated from whole genome data, affected the probability of a locus to be highly differentiated within islands (1% $F_{ST}$ outlier loci; Fig L2 in S1 Text). Examining outliers with extreme values of LLAP only, we found that outliers within Sicily more frequently have excess of Spanish ancestry compared to the genome wide expectation (mean frequency proportion based on 10000 random draws = 0.31), whereas Corsica and Crete outliers display an excess of house ancestry (mean frequency proportion based 10000 random draws = 0.804 and 0.799, respectively; Fig L2 in S1 Text). Even though comparison with proportions from a similar resampling analysis for non-outlier loci shows that the excess ancestry is higher than expected given the genome wide levels of ancestry (Corsica: t = 27.175, Crete: t = -30.846, Sicily: t = -4.369, all $P$<1.25e-05), the ancestry pattern of outliers is generally very similar to that of the general genomic background (Fig L2 in S1 Text).

Finally we evaluated whether genomic blocks with minor-parent-ancestry were more frequently found in regions with high recombination rate, and if such relation has an effect on genomic differentiation, by performing genome-wide correlations between recombination rate and per locus local ancestry proportion (LLAP). Overall recombination rate only explains a small proportion of the variation on minor-ancestry frequency (adjusted R2 < 0.008, Table K in S1 Text). There is a significant positive relationship between minor-ancestry and recombination rate in the Corsica population (whole genome Pearson's correlation: adjusted R2 < 0.0006, $P$ < 2.2e-16), implying that there are more minor-ancestry blocks in higher recombination regions.

For Crete and Sicily the results are inconclusive, while the variation explained is low (R2: 8.0e-05 and 0.008, respectively) the relationships appears to be negative (Fig M and Table K in S1 Text), contrary to what would be expected if there is a higher rate of purging minor parent ancestry in low recombination regions than in regions of high recombination.

## Discussion

While evidence for a creative role of hybridization in evolution is piling up, little is known about how the genomes of hybrid taxa can freely differentiate in response to local selection pressures. Investigating the factors that explain how hybrid taxa can differentiate within lineages of the Italian sparrow, we find surprisingly high genomic differentiation among populations within islands, separated by relatively short distances in light of the dispersal ability of the species [62]. A discriminant function analysis classifies 75–95% of the individuals to the correct population within islands. This local differentiation suggests that there is potential for adaptive divergence within this hybrid species. However, there is more pronounced differentiation among islands, approximately five times higher than *within*-island differentiation, as expected from populations isolated by strong physical barriers.

Interestingly, we find a weak albeit significant correlation between genomic differentiation and a measure of divergent selective sweeps between islands in two out of three comparisons of the islands pairs. However, contrary to our expectation of positive correlations between signatures of divergent selection and genomic differentiation (Fig 1B), we found a weak negative relationship. This does not support a scenario where divergent parental alleles are fixed in response to divergent natural selection. Initial genome stabilization processes, determining admixture proportions, may have been more important than divergent ecological selection. For instance, purging of parental incompatibilities during genome stabilization may have limited the variation for selection to act upon. However, the unexpected direction of the relationship might also reflect that the effects of selection are weak, as selective sweep statistics only explain a very small proportion of the variation in genomic differentiation among islands. Differentiation within islands, likely to have arisen after initial genome stabilization resulting in the island specific admixture proportions, is poorly explained by signatures of selective sweeps. However, within Sicily, haplotype homozygosity (iHS) is weakly correlated with local genomic differentiation. While the weak patterns found in this study offer little support for an important role for divergent selection in population diversification, previous findings are consistent with a role for selection in population differentiation in the Italian sparrow. For instance, local differences in beak shape are best explained by climate and diet for island populations [56]. On the Italian peninsula, population variation in beak shape is best explained by precipitation and genomic differentiation is best explained by temperature [38,55]. These findings are consistent with the large body of work suggesting that hybridization provides the variation facilitating adaptive variation across a range of taxa [2,63–65]. The extent to which signals of selection may be confounded by historical selection acting in the parent lineages, or more recent selection occurring on the hybrid and whether the time frame of hybridization is too short for haplotype based signals to develop, is however not known. However, genomic regions identified as being under selection in the hybrid lineages, using haplotype-based tests, are similar as those previously detected in the house sparrows using whole genome data [61]. This could suggest that observed signals of selection reflect historical selection pressures, but does not exclude the possibility that additional contemporary selection is also reflected in these signatures.

Differences in the degree of genome stabilization can also influence patterns of differentiation among the islands, as purging of incompatibilities and stochastic fixations of parental alleles affect the composition of hybrid genomes. A vast majority of Italian sparrows have house sparrow mitochondrial genome, and among regions that are fixed for house sparrow ancestry across all island populations of Italian sparrow, an excess of nuclear regions with mitochondrial function have been identified [22,23]. This suggests that there has been stabilization of at least parts of the genomes of these hybrid lineages. We find additional evidence suggesting differences in the degree of genome stabilization among the islands. Overall fixation

rates as well as fixation of loci from the major-ancestry parent varied across the islands. Crete has the highest fixation rate, with an elevated fixation of house sparrow alleles follow by Sicily that has a higher fixation rate of Spanish sparrow alleles. Corsica presents the lowest overall fixation rates and did not have differentially fixed alleles from either of the parent species.

We find some evidence suggesting that the same genomic regions repeatedly are involved in population divergence. Differentiation within Corsica is significantly correlated to that within Sicily, but differentiation within Crete is not correlated to that of the other islands. Although our analyses may lack statistical power to detect such correlations, this could also be due to the contribution of *P. domesticus biblicus*, a house sparrow subspecies distributed across the Middle-East, to the population on Crete. This introgression may also have contributed to Crete forming a third cluster in the Admixture analysis. In addition, Corsica shares a higher proportion of the outlier loci than expected by chance with both Crete and Sicily, while the proportion of outliers shared between Crete and Sicily is not higher than expected by chance. These results may, to some degree, support the hypothesis that loci involved in differentiation may be limited to specific genomic regions and are reused across hybrid lineages. Differentiation within island populations could occur in genomic regions that are not under strong selection to fix alleles that are divergent between the parents, after initial genome stabilization where major incompatibilities are sorted, as these regions are likely to be under less strong negative selection.

Measures indicative of selection are consistent across populations and correlated between islands. Hence, the findings of some degree of shared differentiation could partially be explained by similar selection landscapes for all populations of this hybrid taxon or by specific allelic combinations available to selection. Earlier work has shown that the same genetic composition as in the wild ancestor repeatedly arise in lab-crosses of *Helianthus* sunflowers [46] and in younger and older lineages of Lycaides butterflies [47]. It remains to be investigated to what extent the similarity in selection landscapes is caused by historically shared selection in the ancestral populations of the parental species, selection for a functional admixed genome [22,30], stabilizing selection linked to human commensalism [61] or parallel selection for adaptation to insularity. A shared ancestral selection landscape could lead to bias in which parental alleles are retained or more prone to be lost or selected against. For instance, the Spanish sparrow is not considered commensal across most of its range, whereas the Italian and the house sparrow share a commensal ecology. Potentially resulting in consistent selection for specific house sparrow alleles in the independent island lineages of Italian sparrow. We also find variation in the strength of correlation of differentiation among subpopulations comparisons. This pattern is consistent with some degree of parallelism in selection rather than background selection, as we would expect differentiation to be correlated across all comparisons in case background selection strongly limits which areas of the genome are free to vary. If parallel selection is pronounced, we would instead expect the relationship between genomic differentiations to be stronger in some pairs than in the other pair-wise comparisons, and hence variation in the strength of correlations as observed. However, neither of these forces strongly affected the distribution of differentiation, as none of the comparisons were significant when correcting for multiple testing.

A major finding is the limited evidence for genome structure in shaping local differentiation. Variation in the underlying recombination rate landscape may mould the landscape of differentiation [66]. It has been shown to affect the genomic differentiation, generating correlated patterns of differentiation in divergent populations of mice, rabbits [51], flycatchers [48], stonechats [67] and warblers [68], among others. In admixed lineages, selection against minor parent ancestry has been hypothesised to generate patterns of strong correlation between measures of introgression and recombination rate [20,49,52]. This type of selection might be

expected to reduce the genetic variation available for differentiation among hybrid populations. As only the latter process is hybrid specific, a decoupling of the correlation between recombination rate and differentiation present among the parent species is expected if purging of minor parent alleles is important in hybrid taxa (Fig 1A). Recombination rate only explains a small fraction of the variation of minor-parent ancestry proportion. While we find a weak positive relationship between recombination rate and minor-parent-ancestry proportion in Corsica, the pattern is reverse in the other two islands. However, an interesting finding is the stepper correlation between differentiation and recombination rate for the parent species than that among Italian sparrows within islands (Fig 4A). This could suggest that selection against incompatible minor parent alleles in low recombination regions reduces the potential for differentiation in these regions within the hybrid species. However, overall very little of the differentiation within islands is explained by recombination rate, despite the observation of a weak correlation in Sicily. Furthermore, recombination rate overall did not significantly improve models explaining within island differentiation.

Differentiation between the parent species could potentially affect the diversity available for adaptation in the hybrid (Fig 1C), as sorting of ancestry blocks during the genome stabilization process could lead to either fixation of a single ancestry across the hybrid lineage, or of alternative parental blocks in independent hybrid populations. In regions of low parental divergence a lower number of segregating alleles for selection to act on is expected, especially if within-parent diversity is low. A higher evolutionary potential for more divergent loci would be consistent with findings that hybrids from more divergent parent species are morphologically more novel [69,70]. Nevertheless, dominance patterns could also affect the resulting phenotypes in early generation of hybrids [71]. On the other hand, genomic regions of high divergence between parental species can harbour potential genomic incompatibilities in the hybrid taxon. This could generate a negative relationship between the genomic differentiation in the hybrid populations and highly divergent parental loci as the hybrid can only fix ancestry from one of the parent species (Fig 1C). Our data does not lend support to any of these predictions, as we find that overall differentiation between the parent species explains neither the degree of differentiation within islands, nor improves the fit of the models evaluating differentiation within islands (logistic and GLM-models). This could partly reflect the high levels of polymorphism segregating in both parent species and low levels of fixed differences between parent species in this data set. However, highly differentiated loci across populations of the Italian sparrow in mainland Italy have previously been found to present low parental differentiation [38], suggesting that constraints might have played an important role during the stabilization of the hybrid genome, limiting the variation available to selection. Finally, an additional source of variation in hybrid species could stem from variants that segregate within the individual parental species, but we found no evidence for within-parent differentiation affecting differentiation within the hybrid species.

Whether ancestry is a determining factor for how genomic differentiation is distributed in the hybrid genome is not easily disentangled. The divergence in ancestry proportion from the minor–major parent among island populations of the Italian sparrow [22] enables us to test whether differences in ancestry has affected population differentiation after establishing the admixture proportions during early stages of genome stabilization. Purging of genomic incompatibilities, in form of minor parent ancestry blocks, plays an important role in determining the genetic variation in the Italian sparrow [23,24]. A range of studies has suggested that the probability of retaining neutral ancestry is higher in genomic regions with a high recombination rate [20,41,49,52]. To address if minor parent ancestry, in spite of selection against incompatibilities, could be involved in adaptation within the Italian sparrow, we investigated whether minor parent ancestry was important for differentiation. We did not find any

clear effect of ancestry in population differentiation within islands, as highly differentiated outlier loci were not found in minor-parental ancestry blocks more frequently than expected by chance. However, we found significant correlations between local differentiation and the differentiation to the minor-ancestry-parent for two out of three islands, but with opposing signs. Overall $F_{ST}$ outliers are also more differentiated from the minor-parent blocks than expected based on genome-wide levels of differentiation. This suggests that alleles from the minor ancestry parent segregate at loci that are strongly differentiated within islands. As our findings are mixed this would be interesting to investigate further with e.g. whole genome data.

Recombination rate can determine how ancestry is distributed across the hybrid genome [20,49] and may affect the effect that ancestry has on genomic differentiation. The probability for minor-ancestry blocks to rapidly decouple from potential incompatibilities with the major-parent genetic background increases with recombination rate [20]. This affects how easily regions with minor parent ancestry are retained in low recombination areas, and hence affects the variation available for selection that can fuel divergence between hybrid lineages. Interestingly, the probability of being among the 1% most differentiated loci is best explained by how differentiated a given island population is to the house sparrow. Neither recombination rate, the ancestry of the region, nor the differentiation to the Spanish sparrow significantly affected the degree of differentiation or the probability that the locus was an outlier. This is an interesting finding as Runemark *et al.*, [22] previously also found a bias towards house sparrow ancestry in loci consistently inherited from one parent species across island populations. Specifically, they identified an enrichment of mito-nuclear loci and loci involved in DNA-repair. Potentially, these findings could be indicative of some constraints on differentiation from the house sparrow, as most Italian sparrows are fixed for house sparrow mitochondrial haplotypes [5,22]. Another factor that could contribute to this pattern is the overall lower nucleotide diversity and population size [5,38,61] of the Spanish sparrow that could be consistent with a higher incidence of fixation of mildly deleterious alleles.

## Conclusion

Taken together, our findings of correlated differentiation patterns among islands and sharing of outlier loci as well as similar selection pressures signatures within islands may suggest that similarity in selection pressures and/or constraints can contribute to parallelism in genome evolution in the hybrid Italian sparrow. Interestingly, we find that the negative relationship between recombination rate and differentiation expected due to linked selection, being stronger in low recombination regions, was significantly stronger in the parent-parent comparison than within the three hybrid lineages. This could be consistent with a lower differentiation in low recombination regions within the hybrid lineages, as expected if purging of minor parent alleles reduces the variation available for divergence. However, a logistic model revealed that differentiation to the house sparrow is the overall best predictor of the probability of outlier status. Jointly, this suggests that selection interacts with constraints linked to admixture during the stabilization of hybrid genomes.

## Materials and methods

### Ethics statement

All relevant sampling permits were obtained from the regional authorities and handling of birds was conducted according to their guidelines. (Museum National d'Histoire Naturelle, Centre de Recherches sur la Biologie de Populations d'Oiseaux, Paris (France), Institute for Environmental Protection and Research–ISPRA (Italy)–Prot 11177, 23557, Consejería de Industria, Energía y Medio Ambiente (Spain), Norwegian Food Safety Authority (Norway),

Bundesamt für Umwelt BAFU, Abteilung (Switzerland)) and Ministry of Education and Science (Republic of Kazakhstan). Permits approval was granted by the above named boards in the corresponding country of sampling.

## Background

The Italian sparrow originated from hybridization between the house and Spanish sparrow, likely during the spread of the commensal house sparrow to Europe in the wake of the introduction of agriculture [61,72]. The parental species diverged approximately 0.68 million years ago [61]. In addition to the distribution on the Italian peninsula, Italian sparrow populations are also found on some Mediterranean islands. These insular populations have strongly differentiated genomes, with different contributions from each parent species [22], and exhibit phenotypic divergence with island specific beak shape matching local temperature and diet [56]. Furthermore, the island populations are evolutionarily independent and are hypothesized to have arisen from individual hybridization events [22]. Runemark *et al.*, [22] show low concordance (pairwise correlations between islands) of $f_d$ statistic [73] across the same windows along the genome, as well as significant differences in ancestry tract lengths between islands, suggesting that the islands populations have evolved independently. These approaches have previously been used to suggest that a single ancient hybridization event resulted in differential lineages of cichlid fishes [2].

## Sampling and sequencing

Three populations of Italians sparrows were sampled from each of the islands, Sicily (n = 76), Crete (n = 77) and Corsica (n = 70) in March-June 2013 (Fig 2A and S1 Table). On each island we sampled individuals from three geographically separated populations (Figs 2B and 3C). Population sample size varied between 16 and 30 (S1 Table). We sampled reference house sparrow parent populations from Norway (n = 11), and Spanish sparrows from Kazakhstan (n = 10). To increase the number of sampled individuals from the parent species, for analyses that work better with approximately equal sample sizes of all taxa, we added house sparrow samples from Switzerland (n = 17) and France (n = 18) and Spanish sparrow samples from the Gargano peninsula (n = 14) and Spain (n = 23); (S1 Table). All birds were caught using mist nests, and blood was sampled from the brachial vein and stored in Queen's lysis buffer. All necessary permits were obtained from relevant local authorities prior to sampling. DNA was extracted using the Qiagen DNeasy Blood and Tissue Kit, (Qiagen N.V., Venlo, The Netherlands) and the product was stored in Qiagen's buffer EB prior to sequencing. We used a RAD-tag approach; library preparation, sequencing, de-multiplexing and removal of adapters were done by Ecogenics GmbH (Balgach, Switzerland; www.ecogenics.ch). Specifically, the restriction enzymes EcoRi and MseI were used for double digest restriction-site associated DNA sequencing (ddRAD). Fragments between 500-600bp were selected with gel electrophoresis and then sequenced using an Illumina Nextseq500 machine with a 1x75bp read sequencing format.

## Data processing and variant calling

First, the quality of all RAD sequences was checked using FASTQC [74]. Raw reads were filtered using the module process_radtag from the software Stacks [75]. Reads shorter than 73 base pairs were discarded as well as those with an uncalled base. To ensure high confidence-based calls, a Phred quality score of 20 (99% accuracy) was used as threshold across a sliding window fraction of 0.1 of the read length. We used BWA-MEM (v 0.7.8) [76] to map the reads to the house sparrow reference genome [5] using default parameters. We re-aligned indels with GATKs (v 3.7) RealignerTargetCreator and IndelRealigner [77,78] and called the variants using HaplotypeCaller [78]. For a detailed description of the variant calling pipeline, see

Cuevas *et al.*, [38]. We filtered SNPs using Vcftools v. 0.1.14 [79], setting the filter parameters to —max-missing 0.8 (20% missing data allowed), —minDP 10.00, —minGQ 20.00 and —maf 0.02. Using PLINK v. 1.9 [80] we pruned linked sites with an $R^2 > 0.1$, calculated from 100 kb sliding windows and a step of 25 bp. VCF-files containing different set of individuals were generated to suit the different analyses (S2 Table). After filtering VCF files contain between 2224 and 2856 high-quality SNPs and with mean proportion of per individual missing data not larger than 0.13.

## I) Genomic differentiation within- and between islands

We tested the hypothesis that the degree of divergence is significantly higher between islands than within islands, reflecting long periods of independent evolution. To this end, we first illustrated the overall divergence between the islands and populations using a Principal Component Analysis as implemented in glPca() in the R package ADEGENET 2.0 [81]. We also evaluated the level of clustering in the data through estimating the cross-validation error for K = 1 to K = 9, and estimated the probability of each individual belonging to these clusters using ADMIXTURE v.1.3.0 [82]. To illustrate the extent to which the divergence was aligned with the axis of parental differentiation, three parental populations of each species were included in these analyses, resulting in 316 individuals approximately equally distributed across the three species (S2 Table) in a VCF file containing 2224 SNPs (S2 Table). To further illustrate the degree of differentiation within islands, we also performed a Discriminant Analysis of Principal Components (DAPC) within each island to address to which extent the local populations can be correctly classified based on the available variation, we used the dapc() function from ADEGENET 2.0 [81]). To characterize potential variation in genomic diversity we also estimated nucleotide diversity for each population. The analyses were performed in 100kb sliding windows with 25-kb steps using vcftools v. 0.1.14 [79]. For estimates of nucleotide diversity non-variant sites were retained, and we did not filter on minor allele frequency.

We investigated whether the differentiation was stronger between islands than within islands, using two approaches. First, we estimated global $F_{ST}$ among populations within islands, as well as pair-wise $F_{ST}$ among islands in 100kb windows using vcftools v. 0.1.14 [79]. The window size was selected as linkage disequilibrium in sparrows is known to decay within this distance [5], and the windows contained on average 1,5 (SD: ±0.89) SNPs. We used a Monte Carlo permutation paired t-test to investigate if pairwise $F_{ST}$-values were higher among- than within islands.

Second, we used an Analysis of Molecular Variance (AMOVA) to formally address what proportion of genetic variance is explained by differentiation among islands, among local populations within islands, within local populations and variation within individuals. We transformed the VCF to a genlight object, where levels of divergence were defined, using the ADEGENET R-package and ran an AMOVA with the poppr.amova() function from the POPPR R-package [83,84]. We assessed significance by randomization of population assignments using a Monte Carlo test with 1000 permutations implemented in the randtest() function from the ADE4 R-package [85]. Several cut-off of missing-ness per loci were also use (5%, 10% and 20%) with the missingno() function to evaluate the sensitivity of the test (Table L in S1 Text). Loci with high percentages of missing data can disturb the Euclidian distance matrix performed by AMOVA.

## II) The relationship between genomic differentiation and recombination rate

We examined the hypothesis that hybrid genome formation influences the association between differentiation and recombination rate. Our rationale was that if purging of minor ancestry is

stronger in low recombination regions, this reduces diversity in these regions and therefore acts as a constraint on differentiation such that we expect greater differentiation in higher recombination regions where the effect of purging is weaker. Alternatively, if local selective sweeps play a more important role in shaping hybrid genomes, we would expect greater differentiation in low recombination regions [48,51]. As the relative strength of these processes is unknown, we used the relationship between differentiation and recombination rate between the parent species as a null expectation, and tested if there was a deviation from this relationship in the direction expected from purging of minor parent ancestry in the hybrid populations compared to the parent species (see Fig 1A). To this end, we tested for differences in the slopes of individual linear regressions of $F_{ST}$ and recombination rate. We also evaluated a significant interaction between lineage combination (parent-parent vs. within island) and recombination rate on $F_{ST}$ using independent linear model per island. We used recombination rate estimates from Elgvin *et al.*, [5]. We also evaluated the significance of the relation between genomic differentiation and recombination rate within each island using Pearson's correlation tests. We used Bonferroni corrected *P*-values to account for multiple comparisons.

### III) The concordance of patterns of selection and genomic differentiation

To address if elevated genomic differentiation is driven by strong divergent selection, we performed Bonferroni corrected Pearson's correlations of $F_{ST}$ between island pairs to their cross-population Extended Haplotype Homozygosity statistic (xp-EHH) [60], which is designed to compare haplotype lengths between populations (between islands in this case) in order to detect selective sweeps. We further investigated whether regions putatively under selection within-island are independent across islands and whether they coincide with areas of elevated differentiation. We performed pairwise Bonferroni corrected Pearson's correlations between each island pair of the integrated haplotype homozygosity score (iHS) [86] developed for detecting positive selection within a population, in this case calculated within each island. Then, we tested if putative concordance in selection may result in correlated patterns of differentiation in islands, through investigating the correlation between iHS-scores and genomic differentiation for each island. We estimated long range haplotype statistics through phasing data with SHAPEIT/v2.r837 [87,88] and converted the resulting VCF-file using the vcfR R-package [89]. We then used the functions data2haplohh(), ihh2ihs() and ies2xpehh() from the rehh R- package [90,91] to prepare the data, estimate the integrated haplotype homozygosity score (iHS) and estimate Extended Haplotype Homozygosity (XP-EHH), respectively.

### IV) Distribution and repeatability of differentiation across the genome

To test if the differentiation landscape between populations within islands is correlated to that within other islands and between islands, as would be expected if differentiation is affected by the underlying recombination rate landscape and constraints or similar selection pressures acting on the populations, we performed pairwise Pearson's correlation tests on $F_{ST}$ estimates. We tested if global $F_{ST}$ estimates within one island were significantly correlated to these within another island, as well as if between-island differentiation was significantly correlated to global $F_{ST}$ within any of the islands using a resampling approach and Bonferroni corrections for multiple testing.

In addition, we investigated to what extent the same loci were among the most strongly differentiated on different islands. We estimated the proportions of the 1% most differentiated loci that were shared between each island pair. We then investigated if this proportion of shared $F_{ST}$ outliers was higher than expected by chance using a series of $\chi^2$-test for each pairwise comparison, applying Bonferroni corrections for multiple testing. We also provide a list

of candidate genes that are in the vicinity of outliers shared between comparisons. We extracted coding regions within 100kb distance from the shared loci, as linkage decays at approx. 100kb in the house sparrow [5], using the house sparrow annotation file developed by Elgvin et al. (2017).

To further differentiate whether background selection or adaptive parallel selection determine shared patterns of differentiation we run correlations of all possible pair-wise comparison of subpopulations *within-* and *between*-islands. The rationale is that background selection is expected to give rise to correlations in all comparisons as the recombination rate landscape is projected to be constant, whereas parallel selection pressures would generate correlations only in the comparisons where these selection pressures are shared. To correct for multiple testing we performed a resampling approach by running 100 iterations of the correlations.

## V) Patterns of local genomic differentiation in relation to parental contributions to the genome

To evaluate how multiple factors, like genomic parental contribution, parental differentiation and recombination rate among others, may affect which loci are free to vary within the Italian sparrow we performed a generalized linear model (GLM) using within-island $F_{ST}$ as the response variable: $F_{ST}$ = per locus local ancestry proportion (LLAP) + recombination rate + island + island to house sparrow differentiation ($F_{ST}$) + island to Spanish sparrow differentiation ($F_{ST}$) + parental differentiation (house-Spanish $F_{ST}$). We also evaluated how these factors affected the probability of a locus belonging to the 1% most differentiated loci within an island using a similar model with a logistic regression where the response variable was the Pr(outlier). In addition, we performed logistic regressions and GLM individually for each island, excluding the island term. As post hoc tests, we examined Bonferroni corrected Pearson correlations of within-island differentiation against differentiation of the island to each of the parental taxa as well as between the parent species. We also assessed whether highly differentiated loci found in the Italian sparrow are also involved in the genomic differentiation among populations within each parent species (within-house $F_{ST}$ and within-Spanish $F_{ST}$).

We evaluated the degree of genomic stabilization in the different island populations by comparing fixation rates of parentally divergent loci in the Italian sparrow. We also investigated fixation of major- and minor-ancestry parent individually. Loci fixed for different alleles for the two parent species ($F_{ST}$ = 1) were identified from whole genome sequencing (WGS) data for the parental species retrieved from [61] and [5]. For these loci fixation rates were evaluated on WGS data from [22] for Crete, Corsica and Sicily. A total of 17887 SNPs were found to be differentially fixed between parental species and these loci were used to calculated fixation levels in one subpopulation of the Italian sparrow per island.

To address if variation in minor parent ancestry affects within-island differentiation, we tested the correlation between genomic differentiation and the proportion of per locus local ancestry (LLAP) reflecting the relative contribution of each parent species. We estimate a per locus local ancestry proportion (LLAP) using whole genome data from [22], [61] and [5]. To this end we phased data using SHAPEIT/v2.r837 [87,88] and inferred ancestry estimates using LOTER [92]. These were then translated into a per locus local ancestry proportion (LLAP), where values of 0 correspond to loci where only Spanish ancestry is present across all individuals in the population evaluated and 1 corresponds to pure house sparrow ancestry. We estimated the LLAP separately for each island. We also tested if highly differentiated loci were found in blocks with high allele frequencies from major- (greater than 65% major parent alleles) or minor parent ancestry (greater than 65% minor parent alleles) more frequently than expected by chance. This was achieved by comparing the confidence intervals from 10000

resamplings of 8 outlier loci to the value for the entire $F_{ST}$-distribution to assess significance. The same analysis was run for the distribution of non-outlier loci to assess whether the outliers diverge from the neutral expectations.

Finally, we evaluated whether genomic blocks of minor-parent ancestry are more common in regions with high recombination rates, as high recombination rate allows target loci to escape linkage with loci incompatible with the major-parent genomic background. We evaluated to which extent recombination rate explained the proportion of minor parent ancestry through Pearson's correlations between recombination rate estimates retrieved from [5] and the proportion on minor ancestry (LLAP). All data generated in this study can be found in [93].

## Dryad DOI

https://doi.org/10.5061/dryad.wpzgmsbns

## Supporting information

**S1 Table. Sampled individuals from the parent species (the house and Spanish sparrows) and the Italian sparrow.**
(XLSX)

**S2 Table. VCF files.**
(XLSX)

**S3 Table. Genes in the vicinity of shared regions of differentiation within island.**
(XLSX)

**S1 Text. Supporting Figures and Statistics.** It includes: **Table A. Per-island population genomic statistics.** Left panel: Mean values of π and within-island genomic differentiation ($F_{ST}$). Middle panel: t-test for pairwise comparison between genome wide within island $F_{ST}$, evaluating a significance difference between genome wide *within*-island genomic differentiation ($F_{ST}$) across islands. Right panel: Mean values of Tajima's D per population within each island. **Table B. Intercept, slope and confidence intervals of the slope of individual linear regression of *within*-island genomic differentiation and recombination rate as well as parent-parent differentiation and recombination rate. Table C. Evaluating the effect that the interaction between recombination rate and the type of comparison (parental differentiation (house-Spanish), which is the null model, and within-island differentiation) has on genomic differentiation ($F_{ST}$).** Individual linear models per island were run to test if there is a significant interaction between **recombination rate** and **comparison**, as expected if the relationship between recombination rate and differentiation differs between parent species and the hybrid Italian sparrow (Fig 1A). **Table D. Generalized linear model on *within*-island $F_{ST}$. Table E. Logistic regressions per island, on the probability of being a local $F_{ST}$ outlier within island. Table F. Generalized linear models, separated by island on within-island $F_{ST}$. Table G. Concordance of 1.** *between*-island divergent selection (xp-EHH) and **2.** *within*-island selection (iHS) with genomic differentiation ($F_{ST}$). **3.** Correlation between islands of their correspondent *within*-island selection (iHS) estimates. **Table H. Number and percentage of *within*-island $F_{ST}$ outlier loci shared between islands.** Chi-squared denote tests for overrepresentation compared to the genome wide average. **Table I. Number and percentage of *within*-island $F_{ST}$ outlier loci identical to *between*-island outliers.** Chi-squared denote tests for overrepresentation compared to the genome wide average. **Table J. Parallel vs. background selection.** $F_{ST}$ comparisons between within-island subpopulations across all islands. *P*-value, correlations estimates and t-estimates are corrected for multiple testing by

resampling and taking mean estimates after 100 iterations of correlations. **Table K. Linear model of recombination rate and minor-parent ancestry across islands.** The models are performed using values of Log10 of recombination rate as a predictor of local ancestry (LLAP) and dividing these in quartile bins to group the recombination rate values and facilitate interpretation. **Table L. Different cut-offs for the Analysis of Molecular Variance (AMOVA) across islands and populations within islands.** Several cut-offs for missing-ness per loci were used: 5% (see Table 1), A. 10% and B. 20%, but the results from the AMOVA did not change substantially. **Fig A. AMOVA significance—Randomization via permutation.** Monte Carlo test with 1000 permutations implemented in the randtest() function from the ADE4 R-package to evaluate significance. Black line denotes the observed values of Sigma (Variance in each hierarchical level). **Fig B. Concordance of patterns of selection and genomic differentiation.** Correlations of the integrated haplotype homozygosity score (iHS) and genomic differentiation (within-island FST). 1% FST outliers are indicated in coloured dots in contrast to the non-outlier loci, in black. **Fig C. Distribution of Tajima's D per population in each island. Fig D. Correlation of within-islands differentiation across the three Mediterranean islands.** Bonferroni corrections of the p-values are reported. **Fig E. Correlation of within-islands differentiation vs. between-islands divergence.** Adjusted p-values after resampling and Bonferroni corrections. **Fig F. Parallelism of *within*-island pairwise FST**. Pairwise $F_{ST}$ correlations between populations within island "A" to pairwise $F_{ST}$ estimates of populations within island "B", highlighted in green. Significant correlations before correction for multiple testing highlighted in red. Abbreviations of the comparisons are as follow: CORSICA populations: Muratello (Mur), Pianiccia (Pi), Tiuccia (Pi). CRETE populations: Istro (Is), Mithimna (Mi), Perama (Pe). SICILY populations: Cos (Co), Enna (En), Naxos (Na). Thus pair-wise $F_{ST}$ between Muratello vs. Pianiccia is abbreviated as "*Cor_Mur.Ti*". Similarly, pair-wise $F_{ST}$ between Enna vs. Naxos is abbreviated as "*Sic_En.Na*". Estimate values are corrected for multiple testing using a resampling approach (Table J in S1 Text). **Fig G. Parallelism of *between*-island pairwise $F_{ST}$ across all subpopulations.** Correlations of pairwise-$F_{ST}$ between subpopulation a (from island "A") and b (from island "B") and its contrast pairwise-$F_{ST}$ between subpopulation a' (from island "A") and b' (from island "B"). **1.** Correlations between Corsican vs. Sicilian subpopulations. **2.** Corsican vs. Cretan subpopulations and **3.** Sicilian vs. Cretan subpopulations. Populations' name of each island are presented in Fig F. Abbreviations of the comparisons are as follow: As an example, pair-wise $F_{ST}$ between Muratello (from Corsica) vs. Enna (from Sicily) is abbreviated as "*Cor.Mur_Sic.En*". Similarly, pair-wise $F_{ST}$ between Perama (from Crete) vs. Naxos (from Sicily) is abbreviated as "*Cre.Pe_Sic.Na*". **Fig H. Correlation of within-islands differentiation and the parental species.** Adjusted p-values after Bonferroni corrections. **Fig I. Fixation rate of parentally differentiated fixed sites across the islands Italian sparrow populations.** Fixation rate is presented individually by ancestry. Continuity of the y-axis is broken (dashed line) to minimize the size of the figure in order to include the extreme values of the distribution. **Fig J. Correlations of within-island differentiation and within-parent differentiation (within-house or and within-Spanish sparrow).** 1% $F_{ST}$ outliers are indicated in coloured dots in contrast to the non-outlier loci, in black. **Fig K. 1.** Intraspecific genomic differentiation in the parental species for the within-island $F_{ST}$ outlier loci. Dash lines represent the within-parent $F_{ST}$ global mean. **2.** t-tests evaluating whether within-island outlier loci present higher/lower values than expected by chance in the within-parent differentiation. **Fig L. 1.** Relation between within-island $F_{ST}$ and per locus local ancestry proportion (LLAP). Results of linear regression reported. Dashed lines depict the 1% outliers threshold. **2.** Frequency proportion of outlier loci found in regions of mainly house ancestry (0.65<LLAP) and mainly Spanish ancestry (LLAP<0.35) (minor-major parental ancestry). Distribution of 10.000 random resampling draws of 8 outlier loci. **Fig M. Recombination rate**

**v.s proportion of minor-ancestry** (using LLAP, where values of 1 = 100% house ancestry and 0 = 100% Spanish ancestry). Recombination rate is presented in quartiles using whole genome resequencing data retrieved from Ravinet et al (2018), Elgvin et al (2017) and Runemark et al (2018a). Mean and confident intervals of LLAP are shown per recombination rate quantile. Minor ancestors are as follow: Corsica: minor-ancestry from the Spanish sparrow (LAAP = 0 to 0.5), n = 237.523 SNPs. Crete: minor-ancestry Spanish sparrow (LAAP = 0 to 0.5), n = 294.749SNPs and Sicily: minor-ancestry the house sparrow (LAAP = 0.5 to 1), n = 424.739SNPs.
(DOCX)

## Acknowledgments

We thank Jo S. Hermansen and Maria Tesaker for help with fieldwork.

## Author Contributions

**Conceptualization:** Fabrice Eroukhmanoff, Glenn-Peter Sætre, Anna Runemark.

**Data curation:** Angélica Cuevas, Fabrice Eroukhmanoff, Anna Runemark.

**Formal analysis:** Angélica Cuevas, Fabrice Eroukhmanoff, Mark Ravinet, Anna Runemark.

**Funding acquisition:** Fabrice Eroukhmanoff, Glenn-Peter Sætre, Anna Runemark.

**Methodology:** Angélica Cuevas, Fabrice Eroukhmanoff, Mark Ravinet, Anna Runemark.

**Resources:** Angélica Cuevas, Anna Runemark.

**Software:** Angélica Cuevas, Mark Ravinet.

**Supervision:** Fabrice Eroukhmanoff, Anna Runemark.

**Visualization:** Angélica Cuevas.

**Writing – original draft:** Angélica Cuevas, Anna Runemark.

**Writing – review & editing:** Angélica Cuevas, Fabrice Eroukhmanoff, Mark Ravinet, Glenn-Peter Sætre, Anna Runemark.

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
