## [Decision Letter · Decision Letter 0]

5 May 2021

Dear Dr Cuevas,

Thank you very much for submitting your Research Article entitled 'Predictors of genomic differentiation within a hybrid taxon' to PLOS Genetics.

The manuscript was fully evaluated at the editorial level and by independent peer reviewers. The reviewers appreciated the attention to an important problem, but raised some substantial concerns about the current manuscript. Based on the reviews, we will not be able to accept this version of the manuscript, but we would be willing to review a much-revised version. We cannot, of course, promise publication at that time.

If you decide to revise the manuscript for further consideration at PLOS Genetics, please aim to resubmit within the next 60 days, unless it will take extra time to address the concerns of the reviewers, in which case we would appreciate an expected resubmission date by email to plosgenetics@plos.org.

[LINK]

We are sorry that we cannot be more positive about your manuscript at this stage. Please do not hesitate to contact us if you have any concerns or questions.

Yours sincerely,

Alex Buerkle

Associate Editor

PLOS Genetics

Kirsten Bomblies

Section Editor: Evolution

PLOS Genetics

This manuscript has been reviewed carefully by three referees, each of whom notes their interest in this comparison among the multiple products of hybridization. The reviews provide many suggestions for improvement of the presentation in the manuscript, with explicit suggestions for changes and questions about specifics. Additionally, one review raises an important point about the distinction between the process of sorting of ancestry blocks in genome stabilization and additional evolutionary processes that operate thereafter or simultaneously on genetic variants within ancestry blocks. At present, this complexity associated with hybrid speciation is not incorporated.

Overall the reviews are very positive about the manuscript's objectives but raise questions about the analyses that were performed and the interpretation. To the concerns about analysis I will add the concern that concordance of Fst between sampling groups could arise because of unusually intermediate allele frequency at these loci in the ancestors, rather than a shared history of selection, given that much of the variation in Fst is driven by the marginal allele frequency of the minor allele at bivariate sites (see 10.1534/genetics.112.144758).

I encourage the authors to consider these suggestions and encouragement for improvement of the manuscript, while recognizing that addressing some of the points would likely require additional analyses.

Reviewer's Responses to Questions

**Comments to the Authors:**

Reviewer #1: This manuscript describes patterns of genetic differentiation among populations of Italian sparrows. Past work has shown that Italian sparrows likely evolved following hybridization between house and Spanish sparrows. In addition to Italian sparrows in mainland Europe, Italian sparrow populations exist on several islands in the Mediterranean. These island populations are the focus of this manuscript. Specifically, this manuscript documents patterns of genetic differentiation among Italian sparrow populations within islands and between populations on different islands. A suite of analyses are then used to determine whether and to what extent patterns of genetic differentiation can be explained by recombination rate variation, genetic differentiation between the parental species, etc.

I generally liked the overall aims and focus of the paper. In particular, I think comparing multiple outcomes of the genome stabilization process provides insights into how selection and recombination (and drift) shape hybrid genome composition. However, I think more clarity and additional analyses are needed before robust conclusions can drawn from the results in this paper. My biggest concerns are as follows:

1. During genome stabilization, ancestry blocks (segments) fix by drift and selection (with sizes of the blocks determined by the time to fixation and amount of recombination). After this occurs, selection and drift can still occur within ancestry backgrounds, and really the two processes can occur at the same time even. At present, little distinction is made between sorting of ancestry blocks versus evolution with ancestry backgrounds, even after genome stabilization (this is discussed a bit here and there, but it is not as central as it should be). Nor is it clear from this manuscript (perhaps more clear from past work) how much progress has been made towards genome stabilization. I think these issues are critical for interpreting patterns of genetic differentiation in Italian sparrows. For example, tests of selective sweeps in Italian sparrows could pick up signals of past sweeps that occurred in one of the parents if they simply fixed ancestry blocks from that parent. Likewise, tests of selection based on haplotype blocks would presumably be very dependent on whether or not Italian sparrows still have segregating ancestry blocks from the parents in a given region of the genome.

I think this distinction between ancestry sorting during stabilization and selection and drift operating after stabilization needs to be made more clear, including in the conceptual set up (e.g., Figure 1). I think this should then translate into analyses focused on explaining ancestry (e.g., Fst based on ancestry), genetic differentiation within ancestry, and genetic differentiation overall. I realize that hybrid index (which really should be local ancestry I think) is used in the last set of analyses, but I don't think this is sufficient.

2. Related to point 1, it is not really clear from this paper at least how much progress has been made towards genome stabilization and how independent the islands and populations within islands are in terms of progress towards genome stabilization. There are a few mentions that three independent origins are hypothesized, but not much cited or shown to support this. Knowing how much stabilization has progressed and how independently across islands is important for interpreting patterns of genetic differentiation.

3. I am a bit worried that too few SNPs were sequenced for the window-based analyses used here to be appropriate. As best I can tell, 2224 SNPs were used in the analyses (line 494), but maybe that was only true for the Admixture and PCA analyses. I didn't see any other number of SNPs given elsewhere. Assuming 2224 SNPs and assuming a modest genome size for sparrows (e.g., a few hundred Mbs; didn't see the value), this comes out to about 1 or 2 SNPs per 100 kb window. Thus, the windows would mostly be just individual SNPs. And it isn't clear how many of these SNPs are really informative about ancestry. To be clear, I think the number of SNPs is totally sufficient to describe genome-average ancestry as captured by PCA and admixture proportions, and maybe even average ancestry for chromosomes. But I am less convinced they provided detailed information about local ancestry along chromosomes. It appears that additional whole genome data are available (lines 595-596), but were mostly not used here.

Other detailed comments with line numbers:

L75. Also see https://doi.org/10.1038/s41467-020-15641-x, which instead compares ancient and contemporary natural hybrids (and address the role of recombination rate in genome stabilization/ancestry).

L86-90. These sentences would be more clear if put in the context of ancestry sorting versus evolution within ancestry types as noted above.

L127. Here, "potentially originating from independent hybridization events" denotes a fair bit of ambiguity about how likely or unlikely this is. But for most of the paper, it appears that independent origins are assumed.

L139-141. This assumes similar environmental gradients within and among islands, right? If so, maybe make this more explicit.

L152-154. Are these just averages of window-based Fst estimates, or proper, multilocus estimates of overall Fst (i.e., average of ratios or ratio of averages)?

Fig 3A. The legend here needs more information. I suspect the violin plots show the distribution of Fst for windows, but make this clear, including defiing the dashed segments of the vertical lines. And then note that point and bars (presumably) are the mean and 95%CI on the mean.

L156-157. One can get this from the figure, but how much higher? An effect size estimate would be more interesting than just giving results from a null hypothesis significance test.

Table 1. Maybe I am missing something, but the last row of the table suggests that 99.8% of variation within individuals is not significantly different from 0. Is that the correct interpretation?

L196. I don't think it is okay to equate divergent selection with the results of a specific test for selection, in this case xp-EHH. Speaking of correlations between Fst and xp-EHH is fine, just don't call xp-EHH divergent selection. And as noted above, I am concerned that this haplotype-focused tests might be affected in odd ways by (partial) genome stabilization, and also by the low density of SNPs (assuming just 2224 across the genome). Lastly, the very low r2 values for these tests should be highlighted (i.e., yes these metrics are correlated, but Fst explains almost none of the variation in xp-EHH).

L236-237. Again, how much higher? How big is the effect? Without an effect size, P-values don't really tell you much.

L253. How well explained? What is the r2?

L292. First, this should I think be local ancestry not hybrid index. Hybrid index is normally used to refer to average ancestry for an individual, not average at a locus across individuals. Unless the former is what this metric is. Second, one might expect higher differentiation not based on local ancestry per se but z (1-z) (where z is local ancestry), that is, intermediate versus extremes not one parent type versus the other.

L310. Fine, but isn't this just the basic IBD signal one expects to find in almost any system?

L315-316. What might explain the observed pattern? This is another case where ancestry thinking and progress towards genome stabilization could be useful.

L319. Results from one test are not "patterns of selection".

L448-450. Citation to support this? And maybe summarize past evidence along with citation.

L474. I think phred 20 is 99% confidence, not 99.9%

L479. Is `max-missing' allowing for 80% of individuals to be missing data (no genotype)? If so that is really high and should be acknowledged and justified. And define what each parameter is.

L480-481. This could be a bit more clear. Are SNPs with r2 > 0.1 within 25 bp or 100 kb removed?

L487. Capitalize `component' for consistency.

L494. The number of SNPs should be in the Results too. I don't think you want to bury the fact (I think anyway) that the whole paper is based on 2224 SNPs.

L419-526. I think the description here is very clear, more so than in the main text of the Introduction and Results.

L530. How many SNPs on average per window?

Reviewer #2: In their manuscript entitled ‘Predictors of genomic differentiation within a hybrid taxon’ Cuevas et al. sequence a series of populations on three islands of a homoploid hybrid species, the Italian Sparrow, and use various population genomics methods to compare differentiation within and between islands, as well as between hybrids and the two progenitor species to better understand what factors influence the resolution of hybrid genomes. Specifically, they aim to understand whether differentiation (measured by Fst) within hybrids is a factor of recombination, selection, or the major/minor ancestry of that hybrid population. I think this work is both timely and very important for our field, and I agree that the authors have a powerful system to address these questions.

Overall, I quite enjoyed the manuscript and the results are both intriguing and novel, in my opinion. However, there are many parts of the manuscript that I found quite confusing and I’m not entirely sure that I agree with some of the predictions that the authors lay out as uniquely supporting certain evolutionary scenarios. In this light, I have a few suggestions for clarification throughout the manuscript, as well as concerns that the authors may wish to address in revision.

1. This may be picky, but I feel the authors use the terms ‘constraint’ and ‘contingency’ somewhat differently than how I think the majority of evolutionary biologists use these terms. In my mind, constraint is a product of substantial purifying selection constraining a sequence from evolving (and therefore it is conserved across time). The authors seem to be using this term more so describe a scenario in which hybrid genomes resolve in similar ways (e.g. the same parental allele fixes or rises in frequency), potentially due to the fixation of interacting incompatibilities at other loci. To me, there are two very separate processes. Similarly, although to a lesser extent, it seems that the authors use ‘contingency’ to describe scenarios under which regions of the genome in hybrids resolve to look like one parent, and because of epistasis, this influences the probability that loci elsewhere in the genome will also resolve in the same direction (e.g. the fate of an allele at one locus is defined by the fate of alleles at other loci). While I don’t think this is necessarily entirely out of the realm of how researchers of adaptive landscapes and protein evolution might think of evolutionary pathways from one multi-locus genotype to another (for example, how Lewontin might have envisioned this path), this terminology usually refers to the evolutionary order of mutations, not the resolution of pre-existing variation. Again, I think this is a picky point, but I would encourage the authors to be very explicit about how they are using these terms in their introduction. It would then also help to clarify comments like “What is the relative importance of selection vs. constraints in this process?” (lines 22-23), as really both of the processes the authors are referring to are caused by selection (one being potentially genotypic novelty that facilitates novel adaptations, the other being the removal of incompatible combinations).

2. A major goal of this work is to determine if hybrid genomes are more strongly influenced by selection against incompatibilities versus natural selection via adaptations. I think this is a question of high significance for our field, however, I’m not convinced that the predictions outlined in Fig. 1 uniquely support some of the scenarios that the authors suggest they might, and I think there are some underlying assumptions for some of these predictions that are not clearly laid out by the authors. I address these concerns in the context of each prediction/panel.

- Fig. 1A: One potential issue and one very nit-picky comment:

o Larger issue: Because the authors are using differentiation, not minor parent ancestry, I’m not entirely convinced that the reduction in differentiation in hybrid populations with increased recombination is uniquely showing the removal of incompatibilities. In my mind, the reason why this scenario is so plausible in, for example, Schumer et al. 2018 Science, is because regardless of who the minor parent is, minor parent ancestry is much reduced in low recombination regions, and this uniquely supports the removal of incompatibilities (versus say, removal of slightly deleterious variation). In this present analysis, while the authors also have hybrid populations that differ in what parental species is the major/minor contributor, they instead use Fst between hybrid populations, with the expectation that low recombination regions should show less differentiation than differentiation between the parents. I think there are two issues with this interpretation: 1) This type of analysis assumes that incompatibilities are resolved in the same way between hybrid populations. This is probably not unreasonable, because the hybrid populations being compared here are from the same islands and have roughly the same major/minor parental contributions (within islands). However, this is never explicitly stated as an assumption (although it is hinted at throughout the introduction). But, even still, I am not sure that this relationship uniquely supports removal of incompatibilities- which leads to my second concern: 2) this pattern could also be caused by scenarios in which those hybrid populations are adapting to similar habitats, and/or those hybrid populations are purging the same mildly deleterious alleles inherited from one parent. In all cases the same parental ancestry is removed and hybrids will show less differentiation. I might suggest that the authors further test this prediction by looking at minor parental ancestry across recombination bins for all 9 of their populations (as in Schumer et al. 2018. Science. Another recent example of this type of analyses is performed in Calfee et al. 2021. Selective sorting of ancestral introgression in maize and teosinte along an elevational cline. BioRxiv).

o Smaller point: Should the green and purple lines actually cross in this panel? I agree that the slope of the green (hybrid-hybrid) line should be flatter than the purple (parent-parent) line, but under what scenarios do the authors envision that hybrid-hybrid comparisons should show higher differentiation than parent-parent- comparisons (e.g. that the purple and green lines cross)? I suppose this could be the case in scenarios in which some hybrid populations are adapting to a completely novel environment and are therefore experiencing very strong divergent selection (more so than between parents), but otherwise I would expect that the differentiation among hybrid populations is probably capped at the differentiation among parental species.

- Fig. 1B: it is not clear to me why parallel genomic signatures would be expected under scenarios of adaptation both within and between islands, unless the adaptation is parallel within and between islands. Or, are the authors invoking widespread background selection? Because of the sampling, the authors might be able to differentiate these scenarios: If you have a scenario in which island A has populations a’ and a, and island B has populations b’ and b, you might expect parallel genomic differentiation if populations a’ and b’ are both adapting to the same environment (e.g. there is strong parallelism in selection) and this parallelism affects the vast majority of the genome. In which case, you’d expect parallelism between the a vs a’ and b vs b’ contrast, and also between a vs b’ and a’ vs b contrast, but not the others. Under background selection, one might expect all contrasts to have correlated signatures. Have the authors done these types of analyses to contrast evidence for background selection or parallel ecological adaptation? Just one final note on this type of analyses: in my experience, these whole-scale genome correlations in Fst are very common, especially between recently diverged populations, and they are usually due to shared ancestral variation. In my mind, there is still debate as to whether these correlated landscapes are due to shared adaptive evolution or background selection, with evidence on both sides. For example:

Burri et al. 2015. Linked selection and recombination rate variation drive the evolution of the genomic landscape of differentiation across the speciation continuum of Ficedula flycatchers. Genome Research (which the authors already cite)

Stankowski et al. 2019. Widespread selection and gene flow shape the genomic landscape during a radiation of monkeyflowers. PLoS Biology.

Lastly, If the authors think that this parallelism might be caused by replicated differences in ecology, could they expand on this hypothesis in the introduction? Is there some sort of replicated environmental gradient on these islands that might drive these patterns?

- Fig. 1C- Again, the logic of this a bit confusing to me, and I don’t think these patterns unique support a role for incompatibilities: highly differentiated regions between parental species should include incompatibilities (as the authors suggest), but also many regions potentially involved in adaptation. If different hybrid populations are adapting to the same environment (and this happens to mirror an environment experienced by one of the parents), then Fst between hybrids might be reduced at regions of the genome that are highly differentiated among parents. In addition, and as stated above, there’s also no reason to expect that different hybrid populations will resolve all incompatibilities in the same direction, and therefore no reason to expect that regions harboring incompatibilities will show reduced differentiation in different hybrid populations. As the authors know- and as they lay out in their prediction (V), the likelihood of incompatibilities being resolved in the same direction between populations is strongly related to the major ancestry proportion, but I don’t feel this information is well integrated into this figure, and I think this points gets a bit muddled in the intro and discussion.

3. Aspects of the framing of this manuscript seem a bit off to me. For example, framing of adaption in hybrids being facilitated by unique combinations of alleles is not particularly borne out in the authors’ analyses. While there is some evidence of shared outliers in some analyses and in some cases Fst is correlated with other population genomic signals of selection, the authors neither test nor explicitly show that hybrids experience selection for unique combinations of alleles, particularly combinations that involve alleles from both parental species.

4. I think it would be a help to the gentle reader if the authors provided a bit more sign posting and explicit motivation behind their analyses in the results section- especially in sections III, IV, and V. These sections contain a lot of different models that analyze the same data in very subtly different ways, and there are a lot of instances of pairwise correlations between populations at different scale of sampling. Even just a simple sentence or two throughout these sections might greatly help guide the reader to understand why each analysis was performed and what each analyses is telling us (the authors do this in some places already, I just recommend a bit more).

5. The finding that some outliers are shared among comparisons is quite interesting. It might be useful if the authors provided a list of genes within the shared outliers in the supplement, especially if there are any genes of interest. Could the authors also provide an explanation for why outliers may be shared between Sicily and Corsica but not with Crete (whether it be by independent, parallel selection or low level gene flow or some combination). The authors also find that differentiation to the House sparrow genome is negatively correlated with differentiation within islands, suggesting that Spanish sparrow ancestry may be purged in these islands. The authors discuss mito-nuclear co-adaptation, adaptation to human commensalism, and other potential hypotheses of interest in the discussion, but might this pattern also be caused by the Spanish sparrow having a lower effective population size and thus harboring some mildly deleterious alleles, and therefore, these alleles can be recognized and eliminated by selection in a hybrid background?

6. This is more of a minor point, but there are several spots within the manuscript where the authors simply say that two variables are correlated or related, but do not necessarily say that the two variables are negatively related (with the exception of the test statistic being negative). For clarity and to decrease reader confusion, I strongly recommend the authors denote directionality of these relationships when referring to them in text.

Fig. S2/lines 206-209- do the authors think that the lack of correlation between within island Fst and iHS may more strongly point to background selection rather than adaptation and sweeps within islands?

Fig. 5C- are there any outliers shared in all 3 (for both sets of comparisons)? Particularly for the within-island outliers, this might provide evidence for adaptive parallelism. If no, I would denote ‘0’ in the centers of the Venn diagram.

Fig. 4,5,6- can the authors add the correlation coefficient to these scatter plots? It is very hard to tell by looking at them whether the correlations (when they exist) are positive or negative.

Table 2- can the authors list the island term in the top part of this table as well as the Estimated Marginal Means?

Line 24 I think ‘the result’ should be ‘resulting from’

Line 25- “we investigated whether different combinations of parental genomes can further evolve”- I think the authors are more asking how and how much have the hybrid lineages evolved, not if.

line 51- incompatibilities can also be ancestral-derived (if the derived allele has experienced more than one mutation). I’d just say incompatible allelic combinations or similar.

Line 57- these are not a hybrid species, rather just hybrids in a hybrid zone.

Lines 64-66- “These findings suggest that only specific combinations may be viable in a hybrid lineage or only specific portions of the genome are free to vary, potentially resulting in convergent genomic compositions of hybrid lineages.” The wording here is slightly off- under a model of incompatibilities, either parental allelic combination should restore fitness, so this would not necessarily lead to convergence, unless the hybrid populations had a bias in parental ancestry in the same direction.

line 85-86- “Moreover, linked selection is more efficient in low recombination regions” I wouldn’t necessarily say linked selection is more efficient in these regions per se, but rather that linked selection has a greater impact/influences a larger proportion of the genome.

Lines 89-92 in the evidence for selection against introgression in low recombination regions, I would recommend also citing:

Braindvain et al. 2014. Speciation and introgression between Mimulus nasutus and Mimulus guttatus. PLoS Genetics.

As well as a much more recent article:

Nelson et al. 2021. Ancient and recent introgression shape the evolutionary history of pollinator adaptation and speciation in a model monkeyflower radiation (Mimulus section Erythranthe). PLoS Genetics.

Fig. 3B- some of the colored writing in the DAPC is hard to read- especially the yellow ‘Enna’. Could the authors use black, but perhaps color the sampling locals in Fig 3C to match the points in the DAPC?

Line 176- should ‘relation’ be ‘relationship’?

Line 259-260- “For Corsica and Sicily, there is a significant correlation between within-island differentiation and that against the house and Spanish sparrow, respectively”. However, Fig 6 shows correlation between within Corsica and Corsica-Spanish, and within-Sicily and Sicily-House. Should these be flipped?

Lines 311-317: the observation that genomic differentiation within islands is negatively correlated patterns of selective sweeps is not very well explained, except to say this is inconsistent with a model in which standing genetic variation among parents contributes to adaptation in the hybrids (at a genome-wide level). This perhaps seems more consistent with background selection or universal purging of mildly deleterious alleles. Either way, I think the authors could elaborate on this finding.

Lines 371-372: Though see Thompson et al. 2021. Patterns, Predictors, and Consequences of Dominance in Hybrids. American Naturalist.

Line 376-378: “Indeed, we find that differentiation between the parent species explains neither the degree of differentiation within islands, nor improves the fit of the models evaluating within-island differentiation (logistic and GLM-models).” I find this sentence a bit confusing, because the authors predict a negative relationship (although see my above comments), but in this sentence seem to be citing a lack of relationship as evidence for this hypothesis.

Lines 422-423: “Taken together, our findings suggest that hybrid lineages may use their elevated levels of variation for local adaptation” I would avoid this anthropomorphic language.

Line 425: “similar selection pressures” the authors haven’t really shown similar selective pressures (i.e. the agent of natural selection), but rather have shown that differentiation is correlated within and between islands, which may be a sign of parallel ecological adaptation, but also potentially background selection.

Lines 448-450: “Furthermore, the island populations are evolutionarily independent and are hypothesized to have arisen from individual hybridization events.” Can the authors provide a reference for this?

Reviewer #3: In this manuscript (“Predictors of genomic differentiation within a hybrid taxon”), the authors use a unique dataset of the replicated hybrid species the Italian sparrow to understand the consequences of hybridization at the genomic level. They discuss a number of clear hypotheses (Figure 1) outlining the potential roles of recombination rate, selection, and levels of differentiation (both within the hybrid species and between the parental species). I really enjoyed reading this manuscript and think it addresses a topic that is difficult to study in most vertebrate species! It really pushes our understanding of hybridization and genomics forward. The unique sampling approach—something that to my knowledge can’t be replicated in any other bird species—really makes answering their questions possible. I have a few major comments and a larger number of minor comments (sorry!) that I think will help make the manuscript more scientifically sound and easier to read, but overall, I think this is a really cool study and would fit really nicely in PLOS Genetics! (I also think the inclusion of hypothesis/prediction figures is really underdone in the genomics field, so it was really great to see such nice clear predictions presented! And the figures in general were very beautiful!)

MAJOR COMMENTS:

In the introduction, I think you could more clearly separate the background information from the information introducing the Italian sparrow. On Line 100, you basically start discussing in-depth details about the sparrow system, but you’ve not yet officially mentioned that this will be the focus of the study. (There’s also a semi-extensive section from lines 58-62 about them, in a paragraph that’s more generally background information.) It’s a bit jarring going from a general discussion of the recombination rate hypothesis with Figure 1A, to the next paragraph describing the hypotheses for Figures 1B and 1C in the context of the Italian sparrow. I think you should either discuss all of the hypotheses in the context of the system, or all of them more generally. If you choose to do the former, then maybe moving the paragraph that starts on Line 122 up before you start to talk about any hypotheses would be useful. That way you’ve first introduced the system, and THEN go into details about hypotheses.

I think it would be great to explore the way the Crete population falls out in some of these clustering analyses. What happens if you run admixture or the PCA with just the two parental species and the Crete population? Do you get K=2 as the value with the highest support? Does Crete fall out between the two parentals in the PCA as you’d expect for a hybrid between the two? It just seems a little weird that both Corsica and Sicily have patterns more like what you’d expect for a hybrid species, but Crete is an oddball… You kind of touch on this briefly in the discussion and it seems like the explanation for the pattern is a genetically distinct house sparrow subspecies contributing to the Crete population? Given the patterns in Crete are pretty weird and it does sound like something different is going on there, are all three island populations you include really comparable? I don’t really see this as a huge issue, but it does seem like a bit more support that these are all comparable islands would be useful.

You are making so so many different comparisons throughout the paper with the added layer that many of these comparisons are of FST values (which is already a comparison). For the most part I was able to follow what comparisons you were making and why there were important, but I think it would be really helpful as you revise to make sure that you’re being really careful with the word choice you use to describe all the various different components and stick with them throughout. (island, parental, within, among, populations, minor parent, major parent… etc. etc.) I’ve tried to indicate in my specific comments below some places where things were a bit hard to follow.

MINOR COMMENTS:

Line 4. It’s not clear what you mean by “contingencies” here

Line 23. Typo in “a hybrid species result of”

Line 47. Citation 20 is a great reference for your work—and I think makes a lot of sense to include—but I’m not sure the current location/phrasing is the best way to do that. You just finish talking about hybrid species and hybrid speciation, and as swordtails are NOT an example of a hybrid species (more of a hybrid swarm), this might be confusing for readers.

Line 72. Reduced introgression on the sex chromosomes is a pretty common pattern in birds, but the current phrasing makes it seem like the evidence is only present in a few cases…

Line 81. I think this paragraph would be a great place to explicitly discuss Dobzhansky-Muller incompatibilities. They underly a lot of what you discuss in the intro, but you never explicitly describe them. (It would also really nicely flow into the ideas you discuss in the next paragraph!)

Line 130. What do you know about colonization of these islands? Did the hybrid species form first on the mainland and then subsequently colonize each island? Did the parental species colonize the islands and then separately hybridize? Do you have any information available on the timing of hybridization or colonization?

Line 150. Is there a specific place where you mention the ancestry proportions in each island population?

Line 176. “relation” should be “relationship”

Line 195. Link back to your prediction figure here.

Line 196. It would be helpful to explain what xp-EHH is in the text here.

Line 198. Keep consistent how you refer to different island-island comparisons. Elsewhere you use a hyphen, but here you use “and.” There are just so many comparisons you’re making in this paper, it would be really useful if you were extra careful in being consistent with phrasing throughout!

Line 230. Is there a way to talk about this result that is more intuitive for the way the Figure 5D is presented? The figure has each island you’re comparing to grouped together on the x-axis, maybe you can switch the way you talk about the results to be similar.

Line 239. This section makes much more sense to me upon a re-read, but I did find it pretty confusing to wrap my head around on the first read.

Line 251. Change “per island” to “for each island” ?

Line 252-3. You talk about Corsica and Sicily differently here: “between Corsican populations” and “within-Sicily differentiation” but the comparisons are the same I think.

Line 264. Is this result robust to more stringent cutoffs for FST outliers?

Line 265-273. There have been a few points throughout this section where it would be helpful to have an explicit statement about who the minor parent is for each island (e.g., Line 259, 264). I think it would be really helpful to have this info presented much earlier. I found this part of the results to be pretty confusing—I think because there are a lot of comparisons being made, but the expectations for minor/major parent ancestry haven’t really been well-established or discussed yet. I see similar patterns across all of the islands in Figure 6B, but I don’t think that is your expectation and I’m having a hard time squaring the results as I interpret them in the figure and this part of the results.

Line 287. More explicitly introduce the hybrid index here?

Line 306. Typo “in the light of” should be “in light of”

Line 307. It might be helpful to state how dispersal distance compares to island size?

Line 315. Tie back to your prediction figures?

Line 317. Should “island” be plural here?

Line 354. There are other bird examples too, the current phrasing makes it sound like the evidence is kind of limited. E.g., Van Doren et al. 2017 shows a similar pattern, and I think so does some of Darren Irwin’s papers on warblers.

Line 405. Typo “affect the effect”

Line 437. Maybe use “minor parent ancestry” here as you have elsewhere?

Line 450. This seems worthwhile to mention sooner!

Line 465. It would be useful to indicate that you used a RADseq approach in the Results section.

Line 468. What length of reads did you sequence? On what platform?

Line 476. Typo “re-aligned indels realignment”

Line 482. Add the number of SNPs here.

Line 487. Weird capitalization in PCA

Line 493-494. Is this sentence out of place?

Line 496. Was this also in `adegenet`?

Line 517. This is just one sentence in the supplement, right? If so, just move to the main text.

Line 540-542. Move this information to the Results when you talk about xp-EHH

Line 565. As I mentioned earlier, I think it would be great to investigate if more stringer outlier cutoffs lead to similar results or not.

Line 572-576. The info you include here on the GLM might be helpful in interpreting those analyses in the Results; maybe move earlier?

Line 579. Typo “locus to belong to the” should be “locus belonging to the”

Line 591-592. I think there’s an error in the description of major vs. minor parent ancestry?

Line 593. How does the choice of just 8 outlier loci influence the results? How does choosing different numbers influence things?

Line 595. It might be helpful to bring up this information on how the HI is calculated to have it be the second sentence in this paragraph.

COMMENTS ON FIGURES/TABLES:

Figure 1. maybe adding arrows to panel A would help with matching to your description in the legend? More generally, make sure to link back to panels B and C when you’re discussing those hypotheses in the results section.

Figure 3. (A) the y-axis on this plot is pretty confusing…

Figure 4. (A) typo in panel legend (“v.s” should be “vs.”) and in a number of the supplemental tables); (D) would it be useful to include a line in this plot, given the significant correlation?

Figure 5. (A) make sure p-values match exactly what you include in the text for ease of identification of the exact test; maybe add lines to indicate the significant correlations in the two comparisons here? (B) add lines here too to show the significant correlations? (C) left/right indications are switched in the legend

Figure 6. (A) You use the black versus colored points in this figure and in a number of figures in the supplement, but the only place you really mention what the difference between the two points means is in Figure S7. I assume it means the same thing throughout other figures (outlier vs non-outlier SNPs), but it would be nice to have that indicated somewhere in this figure (as it’s the only in-text one). And again, might be useful to add lines to indicate the significant correlations.

Table 1. Several typos in table legend (“were use” should be “were used”; “did not chance” should be “did not change”; “was maintained” should be “were maintained”)

Figure S1. Could probably remove if you wanted, I’m not sure it adds much to the paper (given the results are shown in a table).

Figure S2. Same comment as for Figure S1.

Table S3. It took me a little while to really understand what was going on within each section of this table. I understand not wanting to split it into three separate tables, but some additional guidance on how to read the info in the table might be helpful!

Table S6 and S7. Switch the order? You mention S7 first in the text (Line 188), so maybe make it S6.

Table S8. Maybe change “per island” to “separated by island” in the title?

Table S9. Missing closing parentheses

**Have all data underlying the figures and results presented in the manuscript been provided?**

Reviewer #1: Yes

Reviewer #2: Yes

Reviewer #3: **No: **The authors indicate that data will be put into repositories upon acceptance of the manuscript.

PLOS authors have the option to publish the peer review history of their article (what does this mean?). If published, this will include your full peer review and any attached files.

Reviewer #1: No

Reviewer #2: No

Reviewer #3: No

---

## [Decision Letter · Decision Letter 1]

9 Dec 2021

Dear Dr Cuevas,

Thank you very much for submitting your Research Article entitled 'Predictors of genomic differentiation within a hybrid taxon' to PLOS Genetics.

The manuscript was fully evaluated at the editorial level and by independent peer reviewers. The reviewers appreciated the attention to an important topic but identified some minor concerns that we ask you address in a revised manuscript

We therefore ask you to modify the manuscript according to the review recommendations. Your revisions should address the specific points made by each reviewer.

[LINK]

Yours sincerely,

Alex Buerkle

Associate Editor

PLOS Genetics

Kirsten Bomblies

Section Editor: Evolution

PLOS Genetics

This revised manuscript has been evaluated by two of the original referees. I apologize for the delay as we sought to also obtain a review from the third original referee. The two reports express their appreciation for the revisions to the manuscript and thorough responses to the previous round of review. I concur. One review notes a few copy-editing points related to clarity that I encourage the authors to address. It would be a good idea at this stage for the authors to work through the copy carefully to find any additional usage and typographical errors. Otherwise I believe the authors have done a very thorough job of responding to review and that this manuscript will be a good addition to our growing understanding of the composition of hybrid lineages.

Reviewer's Responses to Questions

**Comments to the Authors:**

Reviewer #2: In their manuscript entitled ‘Predictors of genomic differentiation within a hybrid taxon’ Cuevas et al. sequence a series of populations on three islands of a homoploid hybrid species, the Italian Sparrow, and use various population genomics methods to compare differentiation within and between islands, as well as between hybrids and the two progenitor species to better understand what factors influence the resolution of hybrid genomes. Specifically, they aim to understand whether differentiation (measured by Fst) within hybrids is a factor of recombination, selection, or the major/minor ancestry of that hybrid population. I think this work is both timely and very important for our field, and I agree that the authors have a powerful system to address these questions.

This is the second time that I have reviewed this manuscript, and I applaud the authors for submitting such a thorough revision! I have only a few residual major comments.

First, I think the figure displaying proportion minor ancestry vs recombination bins would be very useful in the supplement.

Second, in Fig 1 A, given that the variation between hybrid populations is merely a subset of variation between parents, it seems like the hybrid-hybrid differentiation should always be lower than differentiation between parents across all recombination rates (incidentally, this is also what the authors find). I would recommend making this minor change to the figure.

Third, I think the references have been shifted- for example, line 494 uses a cichlid reference for warblers, and the reference for dominance and novel trait expression is listed as Andrews (2010) (the reference for FastQC), rather than the appropriate Thompson et al. 2021 (which is listed as two before in the reference list.

Minor comments:

There are also quite a few grammatical errors and typos:

Line 12: should “signature” be “signatures”?

Line 84-86: This sentence is strangely structured- the topic of the sentence is the variation between taxa, but then the example is how loci can vary. I would rephrase to just say something like “The speed of genomic stabilization varies between hybrid taxa and among loci…” or something.

There are several instance of extraneous ‘e.g.’ and ‘i.e.’s that I think add to confusion rather than clarify the writing (for example, lines 89, 101, 256, 440, 492, 536,).

Line 300: should “island” be “islands”?

Line 343: first mention of how Fst outliers are defined (top 1%), but first mention of outliers occurs ~line 234. Could the authors please define outliers the first time they’re mentioned?

Line 359-60: “While Crete and Sicily present a higher ratio of fixed loci from the house and Spanish sparrow, respectively” is not a grammatically correct sentence.

Line 441: Should “vas” be “vast”?

Line 512: this would only be true in regions of low divergence AND low within species diversity…

598: Should this read “across the same windows”?

671-73: Where are these results presented?

Reviewer #3: I commend the authors on the revisions to this manuscript—it’s much improved! I am satisfied with how the authors addressed my comments and the comments from the other reviewers and editor. I particularly appreciated the revisions to the introduction/framing of the paper and the additional analytical checks to confirm the patterns that are being presented. I think the study is really interesting (!) and the unique biology/sampling of the Italian Sparrow system will be of broad interest to the readership of PLOS Genetics.

**Have all data underlying the figures and results presented in the manuscript been provided?**

Reviewer #2: Yes

Reviewer #3: Yes

PLOS authors have the option to publish the peer review history of their article (what does this mean?). If published, this will include your full peer review and any attached files.

Reviewer #2: No

Reviewer #3: No

---

## [Editor Report · Decision Letter 2]

11 Jan 2022

Dear Dr Cuevas,

We are pleased to inform you that your manuscript entitled "Predictors of genomic differentiation within a hybrid taxon" has been editorially accepted for publication in PLOS Genetics. Congratulations!

Yours sincerely,

Alex Buerkle

Associate Editor

PLOS Genetics

Kirsten Bomblies

Section Editor: Evolution

PLOS Genetics

Comments from the reviewers (if applicable):

I appreciate the additional changes to the manuscript and their thorough documentation.

**Data Deposition**

http://datadryad.org/submit?journalID=pgenetics&manu=PGENETICS-D-21-00438R2

**Press Queries**

---

## [Editor Report · Acceptance letter]

4 Feb 2022

PGENETICS-D-21-00438R2 

Predictors of genomic differentiation within a hybrid taxon 

Dear Dr Cuevas, 

We are pleased to inform you that your manuscript entitled "Predictors of genomic differentiation within a hybrid taxon" has been formally accepted for publication in PLOS Genetics! Your manuscript is now with our production department and you will be notified of the publication date in due course.

With kind regards,

Livia Horvath

PLOS Genetics

On behalf of:
